# Structure of mouse protocadherin 15 of the stereocilia tip link in complex with LHFPL5

**Jingpeng Ge[1†], Johannes Elferich[1†], April Goehring[1], Huaying Zhao[2], Peter Schuck[2], Eric Gouaux[1,3]***

[1]Vollum Institute, Oregon Health & Science University, Portland, United States; [2]Laboratory of Cellular Imaging and Macromolecular Biophysics, National Institute of Biomedical Imaging and Bioengineering, National Institutes of Health, Bethesda, United States; [3]Howard Hughes Medical Institute, Oregon Health & Science University, Portland, United States

**Abstract** Hearing and balance involve the transduction of mechanical stimuli into electrical signals by deflection of bundles of stereocilia linked together by protocadherin 15 (PCDH15) and cadherin 23 'tip links'. PCDH15 transduces tip link tension into opening of a mechano-electrical transduction (MET) ion channel. PCDH15 also interacts with LHFPL5, a candidate subunit of the MET channel. Here we illuminate the PCDH15-LHFPL5 structure, showing how the complex is composed of PCDH15 and LHFPL5 subunit pairs related by a 2-fold axis. The extracellular cadherin domains define a mobile tether coupled to a rigid, 2-fold symmetric 'collar' proximal to the membrane bilayer. LHFPL5 forms extensive interactions with the PCDH15 transmembrane helices and stabilizes the overall PCDH15-LHFPL5 assembly. Our studies illuminate the architecture of the PCDH15-LHFPL5 complex, localize mutations associated with deafness, and shed new light on how forces in the PCDH15 tether may be transduced into the stereocilia membrane.
DOI: https://doi.org/10.7554/eLife.38770.001

*For correspondence:
gouauxe@ohsu.edu

†These authors contributed equally to this work

**Competing interests:** The authors declare that no competing interests exist.

## Introduction

The sensing of sound, movement and balance across the vertebrate kingdom originates within hair cells (*Gillespie and Müller, 2009*). In mammals, hair cells within the cochlea mediate our sensation of sound while the detection of movement and our sense of balance originates with hair cells of the utricle and saccule. Hair cells have a characteristic morphology and harbor actin-filled stereocilia 'staircases' in which the stereocilia are coupled by protocadherin 15 (PCDH15) molecules from one stereocilium linked with cadherin 23 (CDH23) molecules from an adjacent stereocilium (*Kazmierczak et al., 2007*). Upon movement of the stereocilia induced by fluctuations of the surrounding fluid, the PCDH15-CDH23 'tip links' trigger the opening of a mechano-electrical transduction (MET) ion channel at the stereocilia tips (*Beurg et al., 2009*). While the molecular composition of the MET channel is enigmatic (*Wu and Müller, 2016*; *Corey and Holt, 2016*), the fundamental role of PCDH15 in coupling the movement of the stereocilia to the gating of the MET channel is unambiguous and conserved throughout vertebrates, such as mammals (*Jaiganesh et al., 2017*), birds (*Ahmed et al., 2006*) and fish (*Maeda et al., 2017*). Moreover, in vertebrates PCDH15 regulates the development and function of stereocilia in hair cells (*Webb et al., 2011*) while its homolog in flies is essential for the microvilli morphology in follicle cells (*D'Alterio et al., 2005*; *Schlichting et al., 2006*). In humans, mutations in *PCDH15* cause Usher syndrome type 1, an inherited deaf-blindness together with vestibular dysfunction and unintelligible speech, as well as

nonsyndromic recessive hearing loss (DFNB23) (*Ahmed et al., 2008*). These findings demonstrate the functional and pathological importance of PCDH15.

PCDH15 belongs to the cadherin superfamily and consists of 11 extracellular cadherin (EC) domains, an extracellular linker (EL) domain, a single transmembrane (TM) helix and a cytoplasmic domain. Three evolutionarily conserved PCDH15 splice isoforms differ in the C-terminal region of the cytoplasmic domains (CD1, CD2 and CD3) (*Ahmed et al., 2008*), but share a common region (CR) following the TM helix (*Figure 1a*). The complex of PCDH15 with CDH23 has been characterized by x-ray crystallography, revealing a 'hand-shake' antiparallel interaction of the two most N-terminal cadherin repeats of both proteins (*Sotomayor et al., 2012*). Immunoprecipitation and two-hybrid experiments suggest that PCDH15 interacts with several likely MET channel components, including transmembrane channel-like protein 1 and 2 (TMC1, TMC2) (*Pan et al., 2013*; *Maeda et al., 2014*; *Beurg et al., 2015*), lipoma HMGIC fusion partner-like 5 (LHFPL5, also known as TMHS) (*Xiong et al., 2012*), and transmembrane inner ear protein (TMIE) (*Zhao et al., 2014*), although none of these studies have isolated stable and biochemically well behaved complexes. Moreover, no structural information is available to illuminate the interactions of PCDH15 and these MET channel components.

The low abundance of hair cells and tip links within these cells limits biochemical and structural studies of the native tip link and its interactions (*Effertz et al., 2015*). We thus screened co-

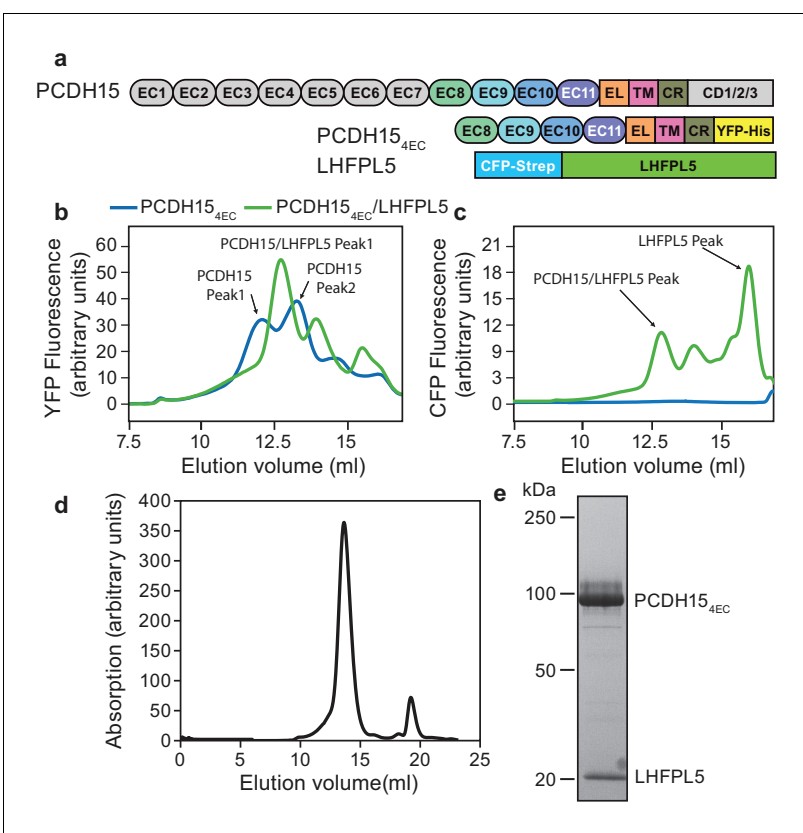

**Figure 1.** PCDH15 and LHFPL5 form a stable and monodisperse complex. (a) Schematic of constructs used for interaction screening of PCDH15 and LHFPL5 by mcFSEC. (b,c) mcFSEC analysis of lysates from cells expressing either PCDH15₄EC alone or together with LHFPL5. Elution of PCDH15₄EC is monitored by YFP fluorescence (b) and elution of LHFPL5 is measured by CFP fluorescence (c). (d) Preparative SEC trace of the PCDH15₄EC/LHFPL5 complex as monitored by absorbance at 280 nm. (e) SDS-PAGE analysis of the PCDH15₄EC/LHFPL5 complex following SEC purification stained using coomassie.

DOI: https://doi.org/10.7554/eLife.38770.002

The following figure supplement is available for figure 1:

**Figure supplement 1.** FSEC analysis of lysates and affinity purification eluates from PCDH15 coexpression screen.
DOI: https://doi.org/10.7554/eLife.38770.003

expression of PCDH15 with TMC1, LHFPL5, or TMIE in a heterologous mammalian expression system and found that PCDH15 forms a robust and monodisperse complex with LHFPL5. We then carried out cryo-EM and x-ray crystallography studies to elucidate the structure of the PCDH15-LHFPL5 complex, providing the first insight into the membrane-integral part of both the cadherin superfamily and the mechanotransduction apparatus..

## Results

### PCDH15 and LHFPL5 form a stable complex

To screen for interaction partners of PCDH15 we created a PCDH15 construct where we deleted seven of the eleven cadherin repeats as well as the C-terminal region that differs between isoforms (PCDH15$_{4EC}$) (*Figure 1a*), arriving at a PCDH15 complex that retained important regions of the extracellular, transmembrane and cytoplasmic domains yet that also possessed monodisperse biochemical behavior. This construct was tagged with mVenus (YFP) and the other putative MET components with mCerulean (CFP) (*Figure 1—figure supplement 1a* and *Table 1*). We expressed PCDH15$_{4EC}$ alone or in combination with LHFPL5, TMC1, or TMIE and monitored the apparent size and monodispersity of the complexes by multicolor fluorescence-detection size-exclusion chromatography (mcFSEC) (*Kawate and Gouaux, 2006*; *Parcej et al., 2013*) in lysates and in pulldowns, under nondenaturing conditions (*Figure 1—figure supplement 1*). Of the three potential subunits, LHFPL5 showed favorable interaction with PCDH15$_{4EC}$. PCDH15$_{4EC}$ alone displayed multiple peaks in the chromatogram, indicating that it was polydisperse when expressed in isolation (*Figure 1b*). Coexpression of LHFPL5 resulted in PCDH15$_{4EC}$ elution as one major peak (*Figure 1b*), a species that persisted following pulldown of the complex (*Figure 1—figure supplement 1b*), in accord with previous observations that PCDH15 and LHFPL5 form a complex (*Beurg et al., 2015*; *Xiong et al., 2012*). LHFPL5 eluted with a peak at the same position as the PCDH15 peak, but also showed a large peak at a later elution volume that likely corresponds to free LHFPL5 (*Figure 1c*). We isolated the PCDH15-LHFPL5 complex by tandem affinity purification, yielding a homogeneous complex that eluted as a single symmetric peak by size exclusion chromatography (*Figure 1d and e*).

### Cryo-EM structure of PCDH15$_{4EC}$-LHFPL5

Imaging of the PCDH15$_{4EC}$-LHFPL5 complex by cryo-EM using a Volta phase plate (*Danev and Baumeister, 2016*) showed particles composed of a micelle-shaped feature with two emerging strands, where the strands adopt a range of conformations (*Figure 2a* and *Figure 2—figure supplement 2a*). For some particles, the strands are almost parallel, with a slight increase in separation toward the tip, distal to the micelle (*Figure 2a*, green). Analysis of these particles, deemed the 'straight' conformation, by 2D classification yields classes consistent with two cadherin chains adopting an approximately parallel configuration (*Figure 2a*, green). We proceeded to carry out a 3D reconstruction and computed a reconstruction at ~11.3 Å resolution of this straight conformation (*Figure 2c*, *Figure 2—figure supplements 1* and *2c and d*, and *Table 2*). The two density features protruding

**Table 1.** Construct information.

| Construct name | Uniprot | Sequence range | N-terminal tag | C-terminal tag |
| --- | --- | --- | --- | --- |
| PCDH15$_{4EC}$ | Q99PJ1 | 1–30,821-1462 | None | mVenus-8xHis |
| PCDH15$_{1EC}$ | Q99PJ1 | 1–30,1145-1462 | None | thrombin-mVenus-8xHis |
| PCDH15 EC11-EL$_{Crys}$ | Q99PJ1 | 1–30,1145-1380 | None | thrombin-mVenus-8xHis |
| PCDH15 EC11-EL$_{AUC}$ | Q99PJ1 | 1–30,1145-1380 | None | 8xHis |
| PCDH15$_{4EC}$ Δ1438 | Q99PJ1 | 1–30,821-1438 | None | thrombin-mVenus-8xHis |
| PCDH15$_{4EC}$ Δ1413 | Q99PJ1 | 1–30,821-1413 | None | thrombin-mVenus-8xHis |
| LHFPL5 | Q4KL25 | 2–219 | Strep-mCerulean-thrombin | None |
| TMIE | Q8K467 | 1–153 | None | thrombin-mCerulean-Strep |
| TMC1 | Q8R4P5 | 2–757 | Strep-mCerulean-thrombin | None |

DOI: https://doi.org/10.7554/eLife.38770.004

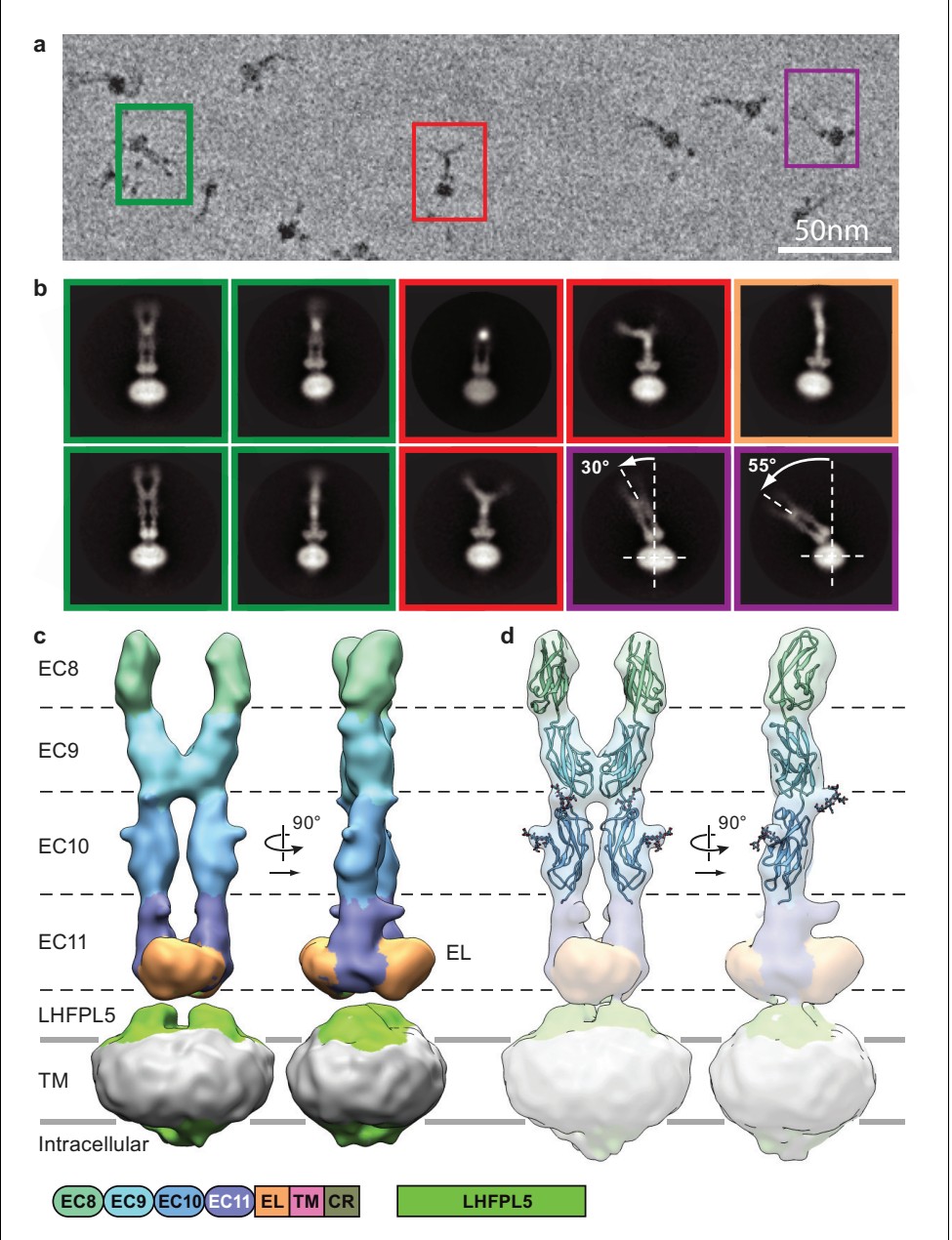

**Figure 2.** Cryo-EM analysis of PCDH15$_{4EC}$/LHFPL5 complex. (a) Representative subsection of a micrograph containing particles of the straight conformation (green), split conformation (red), and tilted conformation (purple). (b) Representative 2D classes. Classes are color coded according to panel (a) with the addition of a bend conformation (gold). (c) 3D reconstruction of the straight conformation. (d) Fit of individual domains from PDB structure 4XHZ into the density. Schematic of the corresponding constructs is listed at the bottom.
DOI: https://doi.org/10.7554/eLife.38770.005

The following figure supplements are available for figure 2:

**Figure supplement 1.** Workflow for 3D reconstruction of PCDH15$_{4EC}$/LHFPL5 complex.
DOI: https://doi.org/10.7554/eLife.38770.006

**Figure supplement 2.** Details of PCDH15$_{4EC}$/LHFPL5 cryo-EM.
DOI: https://doi.org/10.7554/eLife.38770.007

**Table 2.** Cryo-EM data collection, refinement and validation statistics.

| | PCDH15$_{4EC}$/LHFPL5 (EMDB-7327) (PDB 6C13) | PCDH15$_{1EC}$/LHFPL5 (EMDB-7328) (PDB 6C14) |
|---|---|---|
| Data collection and processing | | |
| Microscope | Titan Krios with Volta phase plate | Titan Krios |
| Voltage (kV) | 300 | 300 |
| Electron exposure (e–/Å$^2$) | 27 | 74 |
| Defocus range (μm) | 0.3–1.3 | 0.7–2.2 |
| Pixel size (Å) | 1.72 | 1.04 |
| Symmetry imposed | C2 | C2 |
| Initial particle images (no.) | 288,273 | 972,563 |
| Final particle images (no.) | 16,733 | 78,792 |
| Map resolution (Å) FSC threshold | 11.3 0.143 | 4.5 0.143 |
| Refinement | | |
| Initial model used (PDB code) | 4XHZ | 5B2G, 4P79, 5GJV, 6C10 |
| Model resolution (Å) FSC threshold | 13.8 0.5 | 6.9 0.5 |
| Model resolution range (Å) | | |
| Model composition Non-hydrogen atoms Protein residues | 8505 850 | 6749 862 |
| R.m.s. deviations Bond lengths (Å) Bond angles (°) | 0.003 0.72 | 0.006 1.25 |
| Validation MolProbity score Clashscore Poor rotamers (%) | 1.31 2.32 0.23 | 1.72 5.29 0.27 |
| Ramachandran plot Favored (%) Allowed (%) Disallowed (%) | 95.7 4.1 0.2 | 93.3 6.7 0 |

DOI: https://doi.org/10.7554/eLife.38770.008

from the micelle are ~190 Å in length and ~20 Å in diameter, consistent with an approximately linear chain of four cadherin domains. At the base of the protruding density, proximal to the micelle, is a disk-shaped density that appears to be formed by the C-terminal part of EC11 and the ~100 residue EL domain. The micelle shows two protrusions 'underneath' the disk-shaped EC11-EL density. LHFPL5 contains four α-helices that cross the membrane, intracellular termini, and an extracellular β-strand rich domain that could account for these protrusions. On the intracellular side of the micelle density, there are only minor protrusions that might belong to either PCDH15 or LHFPL5, indicating that the cytoplasmic regions of both proteins are not well ordered.

In a previously reported crystal structure of PCDH15 domains EC8-EC10 (*Araya-Secchi et al., 2016*), there is a ~90° bend between EC9 and EC10, thus giving rise to an L-shaped conformation. We were unable to fit this conformation of EC9-EC10 into our density map and therefore fitted EC8, EC9, and EC10 as independent rigid bodies, guided by the shape complementarity of the rigid-body docked domains (*Figure 2d*, *Figure 2—figure supplement 2f and g*). Furthermore, predicted *N*-linked glycosylation sites map into corresponding protrusions of the density map (*Figure 2d*). We observe a dimer interface between the two cadherin strands formed by the C-terminal half of EC9, largely defined by a loop between β1 and β2 strands and the *N*-terminus of the α-helix between β3 and β4 (*Figure 2—figure supplement 2h*). This interface appears to stabilize the straight conformation.

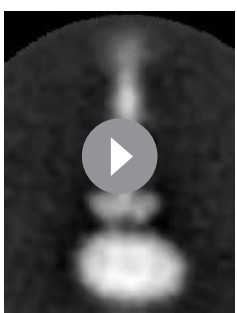

**Video 1.** Animation of 2D class averages of the PCDH15$_{4EC}$/LHFPL5 complex demonstrating the "split" conformation.
DOI: https://doi.org/10.7554/eLife.38770.009

A second group of particles shows the two cadherin chains bent away from one another, deemed the 'split' conformation (*Figure 2a*, red). Inspection of 2D class averages indicates that the cadherin chains are parallel for about half of their lengths, from the micelle, before separating (*Figure 2b*, red and *Video 1*). The density for the most distal cadherin domains is weak, likely due to conformational flexibility, thus precluding 3D reconstruction. Nevertheless, we found that fitting EC10 of the EC8-10 crystal structure into the density for EC10 in our reconstruction produces a conformation that has a striking similarity with 2D class averages (*Figure 2—figure supplement 2i*). This indicates that in our construct the linking polypeptide between EC9 and EC10 can switch between the straight and L conformations. We estimate that the relative populations of the straight and split conformations are approximately equal.

A third class of particles shows a dramatic bending of the entire extracellular domain relative to the micelle (*Figure 2a,b*, purple and *Video 2*). We call this the 'tilted' conformation. The pronounced conformational mobility only occurs in the projection with double-stranded cadherin domains (*Figure 2c*, left), indicating the flexibility is largely within the plane of the cadherin chains. The two 'linkers' connecting the EL domains to the PCDH15 TM helices act as a 'hinge joint', allowing the entire PCDH15 extracellular domains to swivel about an axis parallel to the axis connecting the two linkers, perhaps facilitating a tilted conformation of the PCDH15 extracellular domains upon formation of the mature tip link.

Careful inspection of all 2D classes also shows classes that are combinations of the split and tilted conformations (*Figure 2—figure supplement 2b*). Furthermore, while the cadherin chain dimer appears fairly rigid with respect to the plane formed by the cadherin chains, the chains bend in the perpendicular direction in what seems to be otherwise a straight conformation (*Figure 2b*, gold and *Video 3*). In all likelihood, the split and tilted conformations are not two distinct conformations but rather represent views of a large ensemble of conformational states along a continuous range of motion. The observation that the straight population is sampled frequently enough to allow reconstruction of a 3D structure implies that the straight conformation is stabilized by the EC9 dimer interaction and perhaps by favorable interactions with LHFPL5.

## Crystal structure of PCDH15 EC11-EL

Our understanding of the disk-like structure formed by the EC11 and EL domains in the 3D reconstruction of PCDH15$_{4EC}$ was hindered by the absence of sequence similarity of the EL domain to proteins of known structure. We therefore expressed the EC11-EL construct as a secreted protein from mammalian cells and solved the structure at a resolution of ~1.4 Å by x-ray crystallography (*Figure 3*, *Figure 3—figure supplement 1a* and *Table 3*). As expected, the EC11 domain shows a classic cadherin fold (*Shapiro et al., 1995*), while the EL domain has a ferredoxin-like fold (*Adman et al., 1973*). Between the second β-strand of the ferredoxin motif (β2) and the third strand (β5) resides a β-hairpin (β3-β4) that, in the context of the complex structure, would likely point toward the membrane.

The EC11-EL construct forms a 2-fold symmetric dimer in the crystal lattice (*Figure 3a*). A direct rigid-body fitting of the EC11-EL dimer, derived

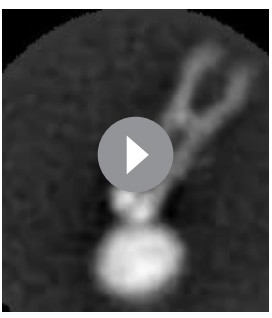

**Video 2.** Animation of 2D class averages of the PCDH15$_{4EC}$/LHFPL5 complex demonstrating the "tilted" conformation.
DOI: https://doi.org/10.7554/eLife.38770.010

from the crystal structure, into the cryo-EM density map of the PCDH15$_{4EC}$-LHFPL5 complex yields a correlation coefficient of 0.94 (*Figure 3c*). The dimer has an oval, disk-like shape with long and short axes of 70 Å and 55 Å, respectively (*Figure 3a*). Unlike the linear 'top to bottom' arrangement of cadherin domains, the EC11 and EL domains sit 'side-by-side', providing extensive interactions within each protomer and throughout the dimer interface. The interactions within each protomer are largely mediated by the β2a and β7 strands in EC11, and the β2 and β5 strands in EL, which together form a β-barrel like structure. A 'major' dimer interface is formed by contacts between EC11 and the EL domain of the symmetry related protomer (EL'). This interface is composed of β4, β5, and β3-β4 loop in EC11, and α1a and β2 in EL' with a buried solvent accessible surface area of ~410 Å$^2$. A 'minor' dimer interface is defined by contacts between EC11 on one protomer and EC11 on the second protomer (EC11'). The minor interface is formed by the guanidino group of Arg1163 and the backbone carbonyl oxygen of Asp1161 (*Figure 3a and b*) burying 140 Å$^2$ of solvent accessible surface area. The C-terminus of the EL domain, which is composed of an α-helix and a loop, extends from the plane of the EC11-EL disk towards the membrane. Because the C-terminal loop in the soluble EC11-EL construct is not connected to a TM domain of PCDH15, its structure likely does not represent the conformation in an intact PCDH15 complex.

The EC11-EL structure displays a striking electrostatic potential distribution (*Figure 3—figure supplement 1c–h*). EC11, with an isoelectric point (pI) of 9.4, mainly shows a positively charged surface, distinct from that of the EC8-10 domains, which have pIs ranging from 4.3 to 4.9. The EL domain has a pI of 5.1 and, accordingly, has a mainly negatively charged surface, except at the C-terminal extended helix, which forms a positive surface together with EC11. A top-down view of the EC11-EL dimer illustrates how residues from α1b and α2b of EL form a negatively charged cavity harboring a deafness mutation, Q1347K (*Ouyang et al., 2005*), at the end of the α2b helix. We note that four conserved arginine residues and one lysine located at the bottom of EC11-EL form a highly positively charged area (*Figure 5—figure supplement 1*). While the function of this basic patch is unknown, it may participate in interactions with other MET channel components.

We also note that there is a strong density in the x-ray maps with a ring-like feature located next to Ser1167, suggestive of O-linked glycosylation. We examined the fit of galactose, mannose and glucose to the density and found that mannose fit best (*Figure 3—figure supplement 1b*). Although the mannose moiety does not directly interact with other residues, it fills the EC11-EL dimer cavity and may help stabilize the dimer.

To measure the propensity of the isolated EC11-EL construct to dimerize, we determined the equilibrium dissociation constant (K$_D$) by analytical ultracentrifugation (AUC). Sedimentation coefficient distribution analysis showed clear transition of monomer to dimer as protein concentration increases from 0.1 to 58 μM (*Figure 3d*) and s$_w$ isoterm analysis using a simple monomer-dimer model yielded a K$_D$ of 5.7 μM (*Figure 3e*). We then attempted to estimate the dissociation rate constant by global analysis of data in three concentrations (*Figure 3—figure supplement 1i–k*). The fitted k$_{off}$ shows a comparatively slow dissociation ($3.5 \times 10^{-4}$ sec$^{-1}$), which is consistent with the expectation from the c(s) overlays. While the micromolar dimer K$_D$ is relatively high, we note that the larger PCDH15$_{4EC}$-LHFPL5 complex appears resistant to dissociation, even at submicromolar concentrations. We thus conclude that the transmembrane regions of the PCDH15-LHFPL5 complex substantially augment the stability of the dimer.

The diffraction of the EC11-EL domain crystals to 1.4 Å resolution, together with the formation of a dimer in solution, shows that the EC11-EL dimer is a robust and well-ordered structural motif. In micrographs of the PCDH15$_{4EC}$/LHFPL5 complex we did not observe any particles that suggest separation of the EC11-EL dimer. In contrast, the dimer interface of EC9 appears to not form in solution (*Araya-Secchi et al., 2016*) and, in the PCDH15$_{4EC}$-LHFPL5 complex, to readily dissociate. The EC11-EL domain, therefore, acts as a robust structural unit, forming a 'collar' that locks the bottom of the filament-like cadherin chains and accepts the pulling force from the cadherin domains. It might be critical to direct the force to the extended helix and loop of the EL domain, which then acts on the PCDH15 TM helix to gate the MET channel.

## Cryo-EM structure of PCDH15$_{1EC}$-LHFPL5

To elucidate the structure of the transmembrane (TM) domains of the PCDH15-LHFPL5 complex we screened additional constructs by mcFSEC, negative stain EM, and cryo-EM. We discovered that a construct containing domains EC11 to CR, named PCDH15$_{1EC}$ hereafter, when coexpressed with full

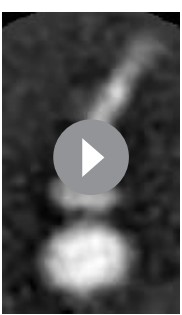

**Video 3.** Animation of 2D class averages of the PCDH15$_{4EC}$/LHFPL5 complex demonstrating the bending of the cadherin chain in the "bend" conformation.

DOI: https://doi.org/10.7554/eLife.38770.011

length LHFPL5 yielded a well behaved complex (*Table 1*). Cryo-EM micrographs showed that this complex yielded near ideal particle distribution and ice thickness, allowing us to use higher magnification and collect micrographs without the use of the Volta phase plate (*Figure 4—figure supplement 2a*). Two-dimensional (2D) classification of the resulting particles showed distinct secondary structure features of the EC11-EL and TM domains (*Figure 4a*, *Figure 4—figure supplement 1b*), indicating that this strategy was promising to result in a higher resolution reconstruction. A 3D reconstruction at ~4.5 Å resolution displayed three layers of density: a top layer formed by the EC11-EL domain, well fitted with the previously described crystal structure; a middle layer formed by the β-strand region of LHFPL5 and the extended C-terminal helix of the

EL domain; and a bottom layer formed by the TM domains of PCDH15 and LHFPL5 (*Figure 4b*, *Figure 4—figure supplements 1* and *2c–e* and *Table 2*). The resulting structure harbors an overall 2-fold symmetry, containing two copies of PCDH15 and LHFPL5. The density for the TM domain containing ten α-helices was well resolved, with some side chain features for bulky residues.

Because LHFPL5 shares unambiguous amino acid sequence homology to claudins (*Longo-Guess et al., 2005*), we rigid-body fitted the claudin crystal structure into the density (PDB: 4P79) (*Suzuki et al., 2014*). We found that two sets of the four 'outer' α-helices of the complex were well fit by the claudin-15 crystal structure (*Figure 4—figure supplement 2f*), indicating that the inner two helices belong to the two copies of PCDH15. We carefully explored alternative fittings of the LHFPL5 model to the density map yet found no better fits, thus demonstrating that the two 'central' helical densities must belong to PCDH15. The two LHFPL5 protomers of the complex interact with one another, via contacts mediated by TM1 helices, which are arranged in a V-shape. The two central helices belonging to the PCDH15 dimer possess an inverted V-shaped architecture and insert into the V-shape formed by LHFPL5. We built an initial structure for the PCDH15 TM helices by rigid-body fitting an ideal α-helix into the density and then optimized the overall models manually and with the help of computational tools, as detailed in the methods section (*Figure 4c*). We were able to model all α-helical regions and most of the β-strands of LHFPL5 with the exception of a short loop between β-strands 1 and 2 as well as a long loop between TM helix 3 and β-stand 5 (*Figure 4c*, *Figure 4—figure supplement 2g and h*). This latter loop contributes a strong density that covers the LHFPL5 β-sheet but is not well enough resolved to trace the peptide chain. While we based the amino acid sequence register on tryptophan residues in LHFPL5 and on prominent features in the TM helix of PCDH15 (*Figure 4—figure supplement 2g*), we will not discuss details of side chain interactions here due to the limited resolution of present reconstruction. We also do not visualize the cytoplasmic C-terminus of PCDH15 that includes the CR region and thus are not able provide a structural model for interactions between the CR region and LHFPL5 found in a previous study (*Xiong et al., 2012*).

Within the TM region there are extensive interactions between PCDH15 and LHFPL5 (*Figure 4d and e*). Each PCDH15 protomer makes more extensive interactions with one LHFPL5 protomer (1100 Å$^2$ buried surface area, BSA) compared to the other (LHFPL5', 270 Å$^2$ BSA). We note that within the TM domains dimerization is primarily mediated by the interface of TM1 from LHFPL5 (487 Å$^2$ BSA) together with the interaction of PCDH15 with TM1 of LHFPL5'. While there is a small interface of interactions between the 2-fold related PCDH15 TM helices (126 Å$^2$ BSA), it likely is only a minor contributor to the stability of the complex.

The extracellular domain of PCDH15 is situated 'on top' of the LHFPL5 β-sheet, but only minor direct interactions are apparent. The closest distances are between the LHFPL5 β3-β4 loop and the β5-β6 loop in EC11, as well as between strand β4 of LHFPL5 and the β3-β4 loop of the EL domain (*Figure 4c*). However, the closest Cα distances are 8.5 Å and 9 Å, respectively, indicating that the interface is likely not extensive. There is also an interaction between the LHFPL5 β-sheet and the

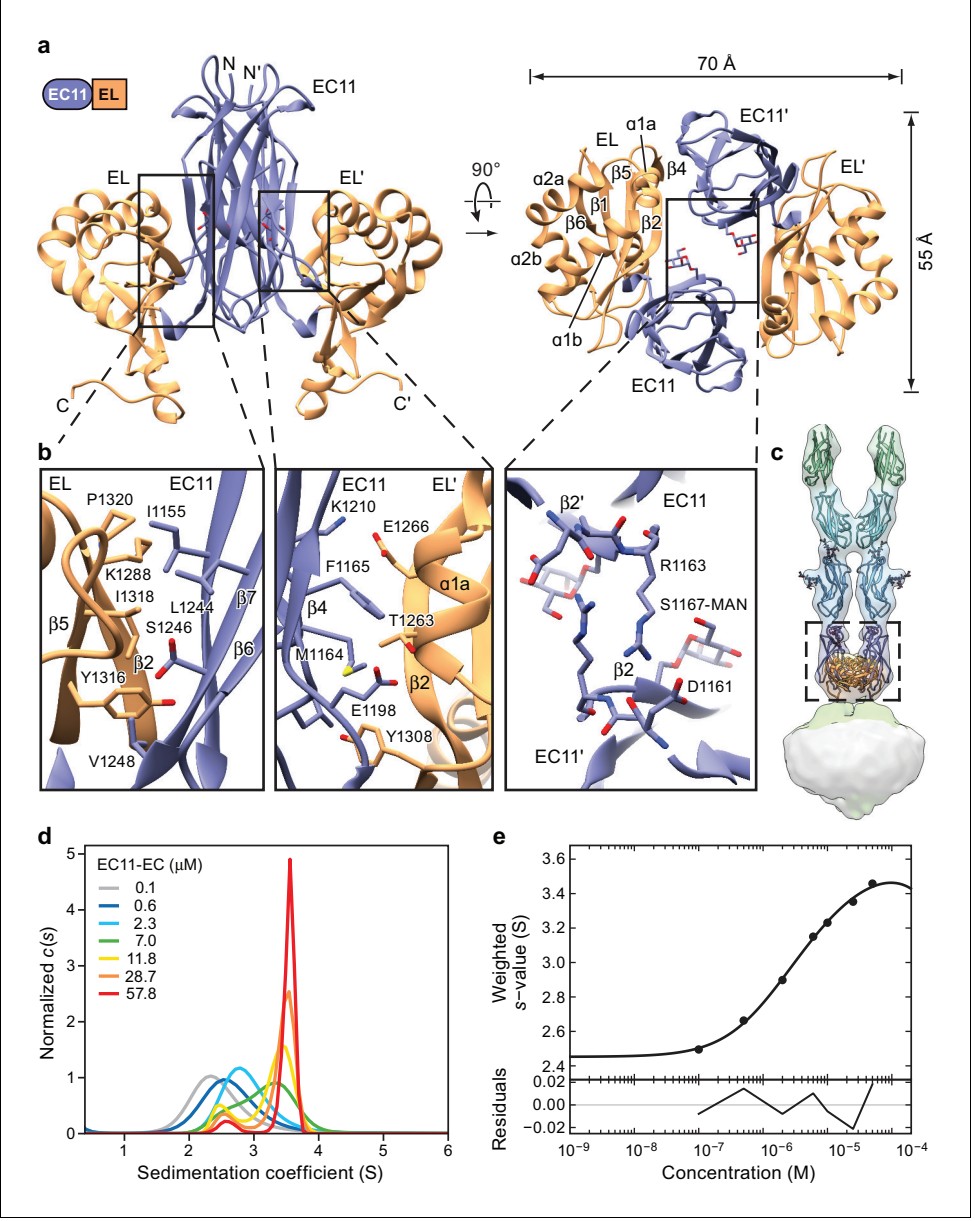

**Figure 3.** Crystal structure of the PCDH15 EC11-EL dimer. (**a**) Ribbon representation of the crystal structure. Mannose modification at Ser1167 is shown in stick representation. (**b**) Detailed view of interactions mediating dimer formation. (**c**) Fit of the PCDH15 EC11-EL crystal structure into the density of the PCDH15$_{4EC}$/LHFPL5 complex. (**d, e**) AUC analysis of the PCDH15 EC11-EL. (**d**) Concentration dependence of the c(s) distributions with loading concentrations between 0.1–57.8 μM. (**e**) Isotherm of signal-weighted average sedimentation coefficient, $s_w$, from the c(s) distributions shown in (**d**). The best-fit value of $K_D$ is 5.7 μM (95% confidence intervals: 3.3–9.9 μM). Best-fit values for monomer and dimer sedimentation coefficients are 2.46 S (95% confidence intervals: 2.37–2.54 S) and 3.74 S (95% confidence intervals: 3.63–3.89 S), respectively.

DOI: https://doi.org/10.7554/eLife.38770.012

The following figure supplement is available for figure 3:

**Figure supplement 1.** Details of PCDH15_EC11-EL crystal structure.

DOI: https://doi.org/10.7554/eLife.38770.013

**Table 3.** X-ray Data collection and refinement statistics

| | EC11-EL |
|---|---|
| **Data collection** | |
| Wavelength (Å) | 0.98 |
| Resolution range (Å) | 54.23–1.40 (1.45–1.40) |
| Space group | P 43 21 2 |
| Unit cell | |
| a, b, c (Å) | 57.68, 57.68, 159.18 |
| A, β, γ (°) | 90, 90, 90 |
| Total reflections | 522785 (35483) |
| Unique reflections | 53914 (5195) |
| Multiplicity | 9.7 (6.8) |
| Completeness (%) | 99.73 (98.04) |
| Mean I/sigma(I) | 25.05 (2.32) |
| Wilson B-factor | 24.1 |
| R-merge | 0.036 (0.52) |
| R-meas | 0.037 (0.57) |
| R-pim | 0.011 (0.21) |
| CC1/2 (%) | 100 (30.7) |
| CC* (%) | 100 (68.5) |
| **Refinement** | |
| Reflections used in refinement | 53914 (5195) |
| Reflections used for R-free | 2696 (260) |
| R-work (%) | 16.9 (33.3) |
| R-free (%) | 19.5 (34.7) |
| CC(work) (%) | 96.5 (57.5) |
| CC(free) (%) | 95.7 (65.0) |
| Number of non-hydrogen atoms | 2141 |
| macromolecules | 1938 |
| ligands | 11 |
| solvent | 192 |
| Protein residues | 244 |
| RMS(bonds) | 0.006 |
| RMS(angles) | 0.75 |
| Ramachandran favored (%) | 98.76 |
| Ramachandran allowed (%) | 1.24 |
| Ramachandran outliers (%) | 0 |
| Rotamer outliers (%) | 0 |
| Clashscore | 2.55 |
| Average B-factor | 39.62 |
| macromolecules | 39 |
| ligands | 68.95 |
| solvent | 44.16 |

*Values in parentheses are for highest-resolution shell.

DOI: https://doi.org/10.7554/eLife.38770.014

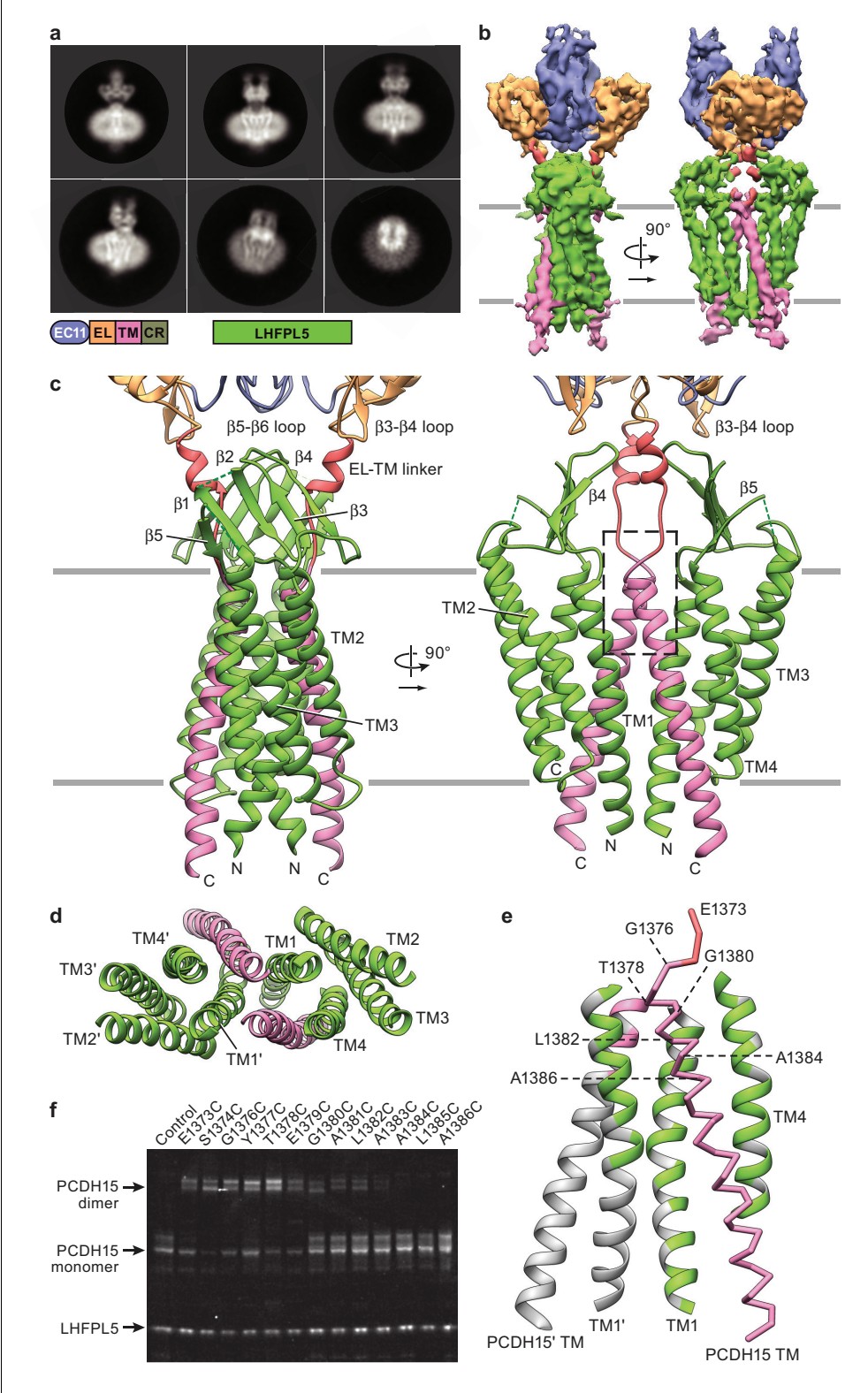

**Figure 4.** Cryo-EM structure of PCDH15$_{1EC}$/LHFPL5 complex. (a) Representative 2D classes. Schematic of the corresponding constructs is shown. (b) 3D reconstruction. (c) Atomic model of the TM portion of the PCDH15$_{1EC}$/LHFPL5 complex. (d) View of PCDH15 and LHFPL5 TM helices from the extracellular side. (e) Interactions formed by the PCDH15 TM helix. One PCDH15 TM helix is shown as a backbone trace with interaction helices shown in cartoon representation, where only residues that potentially interact with the PCDH15 helix are colored. Residues mutated in panel (f) are indicated

*Figure 4 continued on next page*

*Figure 4 continued*

by labels. (**f**) SDS-PAGE analysis of site-directed cysteine crosslinking experiments. PCDH15$_{4EC}$ Δ1413 and LHFPL5 in whole cell lysates expressing indicated mutants are detected using the fused fluorophores.

DOI: https://doi.org/10.7554/eLife.38770.015

The following figure supplement is available for figure 4:

**Figure supplement 2.** Details of PCDH15$_{1EC}$/LHFPL5 cryo-EM.

DOI: https://doi.org/10.7554/eLife.38770.017

helical area of the linker between the EL domain and the TM helix of PCDH15. The density for the PCDH15 linker region is not well defined, indicating that the linker is flexible.

To probe the register of the PCDH15 TM helix, we individually mutated every residue near the extracellular portion of PCDH15 to cysteine and tested for spontaneous formation of a redox dependent dimer by SDS-PAGE (*Figure 4f*, *Figure 4—figure supplement 2h and i*). We found that residues 1373–1379 readily crosslinked, residues 1380–1382 allowed for partial crosslinking, and residues 1383–1386 did not crosslink. The lack of crosslinking of residues 1383–1386 is consistent with our model because these residues are separated by LHFPL5. While the crosslinking of the other residues is consistent with their close proximity in our model, the lack of a clearly defined periodicity indicates that the tip of the helices and the linker region are somewhat flexible. This result, together with the lack of substantial interactions between the extracellular domain and LHFPL5, as well as the weak density for the linker region, paints a picture of an extracellular domain that has the freedom to move substantially with regard to the transmembrane region, as observed in the tilted conformation of the PCDH15$_{4EC}$-LHFPL5 complex structure.

## Complete model of PCDH15$_{4EC}$/LHFPL5 complex

By rigid body fitting the PCDH15$_{1EC}$-LHFPL5 structure into the PCDH15$_{4EC}$-LHFPL5 density we built an almost complete model of the complex (*Figure 5a*). The model shows that the 4 cadherin domains stretch 220 Å above the membrane, with the EC11-EL domain positioned ~27 Å above the TM region (*Figure 5a*). The cadherin domains have a right handed twist of 23.9° from EC11 to EC8 (*Figure 5a,b*), consistent with a right-handed coil of the tip link observed in high-resolution EM imaging (*Kachar et al., 2000*). However, the observed twist is not enough to explain a 60 nm periodicity (*Kachar et al., 2000*), which would require a 120° turn over the length of the ~20 nm cadherin chain. It is possible that the degree of twist is stronger towards the *N*-terminus of the cadherin chain or that the twist is not formed spontaneously, but is instead the result of rotation of either end of the tip link. It is also probable that the observed stiffness is a unique property of the EC11-EC10 pair due to the embedding of EC11 into the EC11-EL collar.

The separation of the center of masses of LHFPL5 and EC8-EC10 is almost identical (~30 Å), with EC8 showing a slightly larger separation (~45 Å). This is caused by a 23° outward tilt of EC9, compared to the other cadherin domains that show substantially smaller tilts (*Figure 5c*). This tilt facilitates the dimer interaction of EC9. While these observations might be skewed by the relatively small proportion of particles in the final reconstruction or by the imposed C2 symmetry, the separation of EC8 and tilt of EC9 is apparent in 2D class averages of the straight as well as the tilted conformation (*Figure 2b*).

A striking feature of our model is the overall geometry of the linker connecting the PCDH15 transmembrane helix with the EL region (*Figure 5a*). The two PCDH15 transmembrane helices come within 7 Å of one another at the extracellular side of the membrane. As the linkers transition toward the extracellular domain they separate, resulting in a distance of 21 Å of the Cα atoms at the *N*-terminal end of the linker α-helix. The main chain then connects to β-strand 6, which extends to the outside of the EL domain, resulting in a final separation of 62 Å. This suggests that a pulling force acting on the cadherin chain would pull the extracellular tips of the PCDH15 transmembrane helices apart. This architecture is consistent with a role of the EC11-EL collar in providing a local geometry of the polypeptide chain that could facilitate conversion of a pulling force into a conformational change that ultimately leads to an opening of the MET channel.

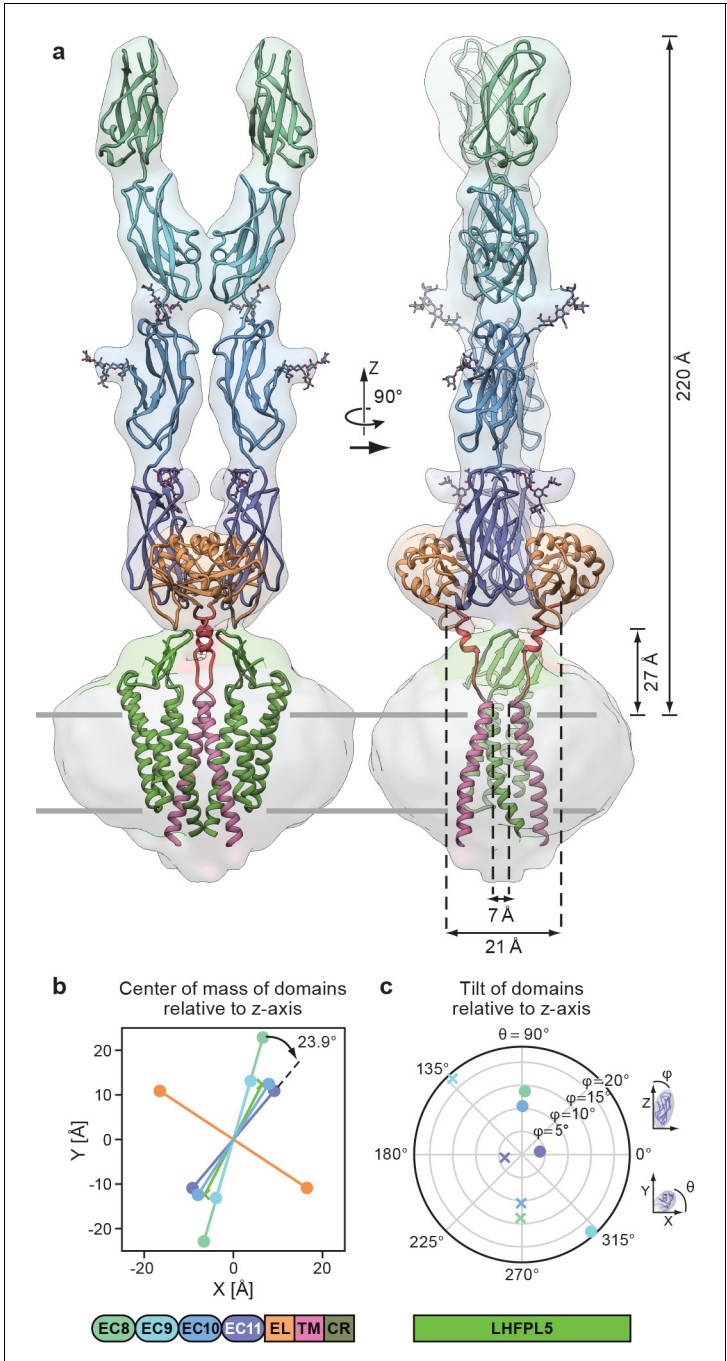

**Figure 5.** Complete model of the PCDH15$_{4EC}$/LHFPL5 complex. (a) Complete model generated from fitting subunits into the PCDH15$_{4EC}$/LHFPL5 map, shown together with the map. Key dimensions are indicated. In the panel on the right hand side one LHFPL5 molecule has been omitted for clarity. (b) Center of mass of individual domains projected along the z axis, which is defined as the symmetry axis of the complex as indicated in panel a. A 23.9° twist of the EC11 compared to EC8 is indicated. (c) Tilt of the principal axes of the cadherin domain compared to the z axis. Domains of chains A and B are shown as dots and crosses, respectively. At the bottom of the figure is a schematic detailing the color coding of the individual domains.
DOI: https://doi.org/10.7554/eLife.38770.018

The following figure supplement is available for figure 5:

**Figure supplement 1.** Multiple sequence alignments of PCDH15$_{4EC}$ (a) and LHFPL5 (b).
DOI: https://doi.org/10.7554/eLife.38770.019

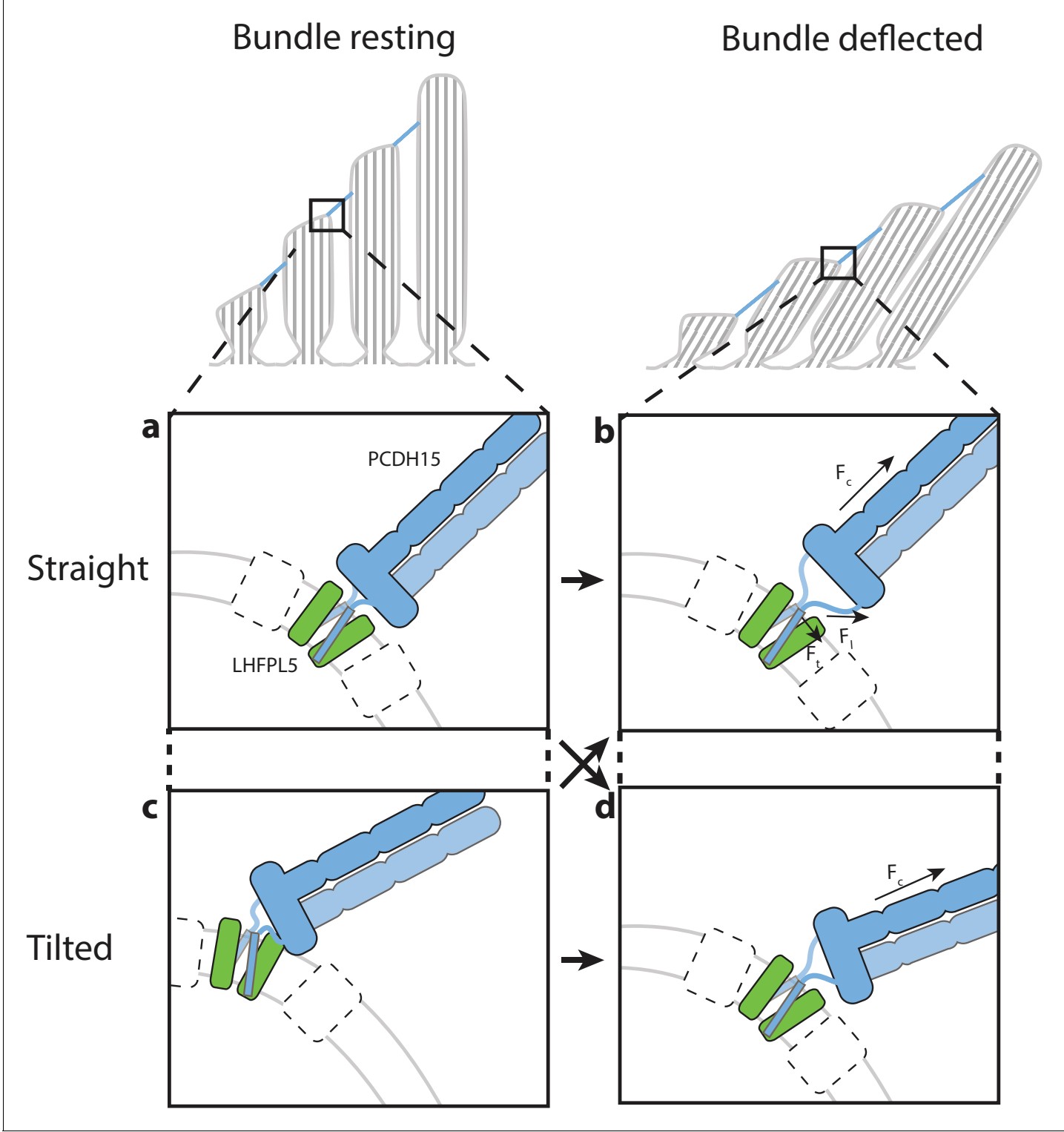

**Figure 6.** Hypothesis for the role of the PCDH15$_{4EC}$/LHFPL5 complex in mechanotransduction. With the hair bundle of a hair-cell at rest the PCDH15 – LHFPL5 complex could either adopt the straight (**a**) or tilted-conformation (**c**), depending on whether the orientation of the membrane surrounding PCDH15 is perpendicular to the tip-link direction. Deflection of the hair bundle applies force to the PCDH15 cadherin domain chain (F$_c$) (**b, d**). This might result in the conversion from the tilted (**c**) to the straight conformation (**b**) or from the straight conformation (**a**) to the tilted conformation (**d**). In the case of the straight conformation (**b**) F$_c$ is converted by the EC11-EL collar to a force on the linkers that, due to the local geometry, is no longer perpendicular to the membrane (F$_l$). We speculate that this could pull the PCDH15 transmembrane helices apart (F$_t$). White boxes with a dashed outline

*Figure 6 continued on next page*

*Figure 6 continued*

represent TMC, TMIE, or other so far unidentified proteins that could bind to either PCDH15 or LHFPL5 and sense this movement of the PCDH15 transmembrane helix or the change in membrane environment, ultimately leading to the opening of the MET channel.

DOI: https://doi.org/10.7554/eLife.38770.020

## Discussion

Here we demonstrate that PCDH15 forms a robust complex with LHFPL5 and that the assembly is composed of 2 copies of each protein, thus providing concrete evidence that the composition of PCDH15 in the tip link is a dimer. The conformationally mobile EC8-EC10 domains are poised to facilitate the search for CDH23 during tip link formation while the membrane proximal EC11-EL domain forms a robust, dimeric collar, which we speculate aids in transduction of force to the membrane-embedded PCDH15 TM helices (*Figure 6*). The PCDH15 TM domains, in turn, are stabilized by extensive interactions with 2 LHFPL5 subunits, interactions that are consistent with LHFPL5 being important for assembly and trafficking of PCDH15 (*Xiong et al., 2012*). Interestingly, deafness mutations in LHFPL5 do not affect the interaction of PCDH15 and LHFPL5 (*Xiong et al., 2012*) and in our structure do not localize to interaction sites of PCDH15 and LHFPL5, suggesting that LHFPL5 interacts with additional components of the mechanotransduction complex (*Beurg et al., 2015*; *Xiong et al., 2012*) (*Figure 6*). Because the material obtained for structure determination is not purified from hair cells we cannot exclude the possibility that the unique cellular architecture, proteome (*Krey et al., 2017*) or lipid composition (*Zhao et al., 2012*) of hair cells could influence the conformation of the complex and the observed interactions between the protein components.

The two linker regions connecting the PCDH15 collar with the TM domains might act as 'ligaments' of a hinge joint, allowing the tilting of the PCDH15 extracellular domains. Therefore, PCDH15 could accommodate a membrane orientation at the lower insertion point that is not perpendicular to the tip link (*Figure 6*). It is also possible that PCDH15 will assume a straight conformation at rest and a tilted conformation under tension, or vice versa. Experiments that measure the angle between the PCDH15 extracellular domain and the membrane as a function of bundle deflection will be needed to address this question. Our observation that the PCDH15 extracellular domain only tilts along one direction could be related to the directional sensitivity of the hair bundle towards deflection (*Shotwell et al., 1981*). We speculate that force transduced through the PCDH15 cadherin domains leads to separation of the PCDH15 TM helices, near the extracellular side of the membrane, and potentially to the conversion between the straight and tilted conformations. These conformational changes could ultimately lead to the gating of the MET ion channel. Taken together, our studies define the architecture, stoichiometry and conformational dynamics of the PCDH15-LHFPL5 complex, thus providing insight into the trafficking, assembly and function of the MET channel of hair cells.

## Materials and methods

### Key resources table

| Reagent type (species) or resource | Designation | Source or reference | Identifiers | Additional information |
|---|---|---|---|---|
| Gene (*Mus musculus*) | PCDH15 | Synthetic | UniProt: Q99PJ1 | |
| Gene (*Mus musculus*) | LHFPL5 | Synthetic | UniProt: Q4KL25 | |
| Gene (*Mus musculus*) | TMIE | Synthetic | UniProt: Q8K467 | |
| Gene (*Mus musculus*) | TMC1 | Synthetic | UniProt: Q8R4P5 | |
| Cell line (*Spodoptera frugiperda*) | Sf9 | ThermoFisher | 12659017 | |

*Continued on next page*

*Continued*

| Reagent type (species) or resource | Designation | Source or reference | Identifiers | Additional information |
|---|---|---|---|---|
| Cell line (*Homo sapiens*) | HEK293 tsa 201 | ATCC | CRL- 11268 RRID:CVCL_1926 | |
| Cell line (*Homo sapiens*) | HEK293S GnTI⁻ | ATCC | CRL- 3022 RRID:CVCL_A785 | |
| Recombinant DNA reagent | pEG BacMam | doi: 10.1038/nprot.2014.173 | | |
| Recombinant DNA reagent | Lipofectamine 2000 reagent | Invitrogen | 11668–027 | |
| Recombinant DNA reagent | Cellfectin II reagent | Invitrogen | 10362–100 | |
| Software, algorithm | Unblur | doi:10.7554/eLife.06980 | | http://grigoriefflab.janelia.org/unblur |
| Software, algorithm | UCSF MOTIONCOR2 | doi:10.1038/nmeth.4193 | | http://msg.ucsf.edu/em/software/motioncor2.html |
| Software, algorithm | Gctf | doi:10.1016/j.jsb.2015.11.003 | | http://www.mrc-lmb.cam.ac.uk/kzhang/ |
| Software, algorithm | cryoSPARC | doi:10.1038/nmeth.4169 | | https://cryosparc.com/ |
| Software, algorithm | RELION-2 | doi: 10.1016/j.jsb.2012.09.006 | RRID:SCR_016274 | http://www2.mrc-lmb.cam.ac.uk/relion |
| Software, algorithm | DoG-picker | doi: 10.1016/j.jsb.2009.01.004 | | http://emg.nysbc.org/redmine/projects/software/wiki/DoGpicker |
| Software, algorithm | Gautomatch | | | http://www.mrc-lmb.cam.ac.uk/kzhang/ |
| Software, algorithm | Bsoft | doi:10.1016/j.jsb.2006.06.006 | | https://lsbr.niams.nih.gov/bsoft/ |
| Software, algorithm | UCSF Chimera | doi:10.1002/jcc.20084 | RRID:SCR_004097 | https://www.cgl.ucsf.edu/chimera |
| Software, algorithm | PHENIX | doi:10.1107/S0907444912001308 | RRID:SCR_014224 | https://www.phenix-online.org |
| Software, algorithm | COOT | doi:10.1107/S0907444904019158 | RRID:SCR_014222 | https://www2.mrc-lmb.cam.ac.uk/personal/pemsley/coot |
| Software, algorithm | Rosetta-CM | doi:10.1016/j.str.2013.08.005 | RRID:SCR_015701 | https://www.rosettacommons.org/docs/latest/application_documentation/structure_prediction/RosettaCM |
| Software, algorithm | GlyProt | doi:10.1093/nar/gki385 | RRID:SCR_001560 | http://glycosciences.de/modeling/glyprot/php/main.php |
| Software, algorithm | CHARMM-Gui | doi:10.1002/jcc.20945 | RRID:SCR_014892 | http://charmm-gui.org/ |
| Software, algorithm | NAMD | doi.org/10.1002/jcc.20289 | RRID:SCR_014894 | http://www.ks.uiuc.edu/Research/namd/ |
| Software, algorithm | MDFF | doi:10.1016/j.str.2008.03.005 | | http://www.ks.uiuc.edu/Research/mdff/ |
| Software, algorithm | PyMOL | Schrodinger LLC | RRID:SCR_000305 | http://www.pymol.org |

## Construct design and cell culture

All constructs use the canonical *Mus musculus* sequences as recorded in the UniProt (*The UniProt Consortium, 2017*) database and were synthesized by Genscript. All constructs were cloned in the pEG BacMam vector under the control of a CMV promoter, allowing expression by transient transfection using plasmids and infection using Baculorvirus produced in Sf9 cells (*Goehring et al., 2014*). Details about the sequence ranges, fluorophores, and affinity tags are listed in *Table 1*.

Sf9 cells (ThermoFisher 12659017) were cultured in sf-900 III SFM medium at 27°C. Adherent HEK293 tsa201 cells (ATCC CRL- 11268) were cultured in DMEM medium supplemented with 10% (v/v) fetal bovine serum at 37°C. HEK293 tsa201 cells (ATCC CRL- 11268) in suspension were cultured in Freestyle 293 expression medium supplemented with 1% (v/v) fetal bovine serum at 37°C. HEK293 GnTI⁻ cells (ATCC CRL- 3022) in suspension were cultured in Freestyle 293 expression medium supplemented with 2% (v/v) fetal bovine serum at 37°C. Cells are routinely tested for mycoplasma contamination using CELLshipper Mycoplasma Detection Kit M-100 from Bionique. All of our cells are mycoplasma free. We have not used any cell lines from the list of commonly misidentified cell lines.

## Multi-color FSEC

Adherent HEK293 tsa201 cells were grown in 6-well plates to about 80% confluency and then transfected with 1 ug of DNA using the PolyJet transfection agent. After 8 hr the media was replaced with media containing 10 mM sodium butyrate and the temperature was reduced to 30°C. Cells were harvested 48 hr after transfection. Cells from one well were lysed using 200 µL lysis buffer (20 mM dodecyl-β-D-maltoside, DDM, 50 mM Tris pH 8.0, 150 mM NaCl, protease inhibitors: 1 mM phenylmethylsulfonyl fluoride, 0.8 µM aprotinin, 2 µg/ml leupeptin and 2 µM pepstatin A) for 1 hr at 4°C and lysates were clarified by ultracentrifugation. 30 µL of the lysate was retained for further analysis and the remainder was incubated with 50 µL StrepTactin resin that was pre-equilibrated with wash buffer (1 mM DDM, 20 mM Tris pH 8.0, 150 mM NaCl). After one hour of incubation the resin was washed three times with 500 uL of wash buffer and protein was eluted using 100 µL wash buffer containing 10 mM desthiobiotin. Lysates and elution were then analyzed using a HPLC system with a Superose 6 Increase 10/300 GL column. Protein elution was monitored with two fluorescence monitors tuned to mCerulean and mVenus fluorescence.

## Expression and purification of the PCDH15 EC11-EL domain for crystallization and AUC analysis

For crystallization, the PCDH15 EC11$_{Crys}$ construct (*Table 1*) was expressed in HEK293S GnTI⁻ cells. Cells at a density of $2.0 \times 10^6$ ml$^{-1}$ were infected with BacMam virus at a multiplicity of infection (M. O.I) of 2. Sodium butyrate was added to cultures 12 hr post-infection and cells were transferred to 30°C. Supernatant was harvested 96 hr post-infection, centrifuged at 4,000 g for 15 min and cell pellets were discarded. Prior to binding to TALON resin, the supernatant was filtered, concentrated and adjusted to pH 7.5 with 50 mM Tris (final concentration). The resin was washed with 10 column volumes (CV) of 20 mM Tris pH 8.0, 150 mM NaCl and 40 mM imidazole. The protein was eluted with buffer containing 20 mM Tris pH 8.0, 150 mM NaCl and 200 mM imidazole, followed by PNGase 1:10 (w/w) and thrombin 1:300 (w/w) digestion at room temperature for 3 hr. The sample was dialyzed against 20 mM Tris pH 8.0, 150 mM NaCl for 12 hr and applied to TALON resin to remove mVenus-8His. The flow through was collected and loaded onto a size-exclusion column (Superdex 75 10/300 GL column) pre-equilibrated with 20 mM Tris 8.0, 150 mM NaCl. Peak fractions were collected and concentrated to 5–8 mg/ml for crystallization.

For AUC analysis the PCDH15 EC11-EL$_{AUC}$ construct was used (*Table 1*). Similar expression and purification steps were carried out except that thrombin digestion and dialysis were not performed. Protein after PNGase 1:10 (w/w) treatment was concentrated and further purified by size-exclusion chromatography (Superdex 75 10/300 GL column) using a buffer composed of 20 mM Tris pH 8.0 and 150 mM NaCl. Peak fractions were collected for AUC analysis.

## Crystallization and structure determination of PCDH15 EC11-EL

The best diffracting crystals were obtained at 20°C using the hanging-drop vapor diffusion method by mixing 1 µL protein and 1 µL reservoir solution containing 0.2 M lithium sulfate, 30% (w/v) PEG 4000, 0.1 M Tris pH 8.5. Crystals were cryo-protected with reservoir solution supplemented with 15–20% (v/v) glycerol. X-ray diffraction data was collected at ALS beamline 8.2.1 and APS beamline 24-ID-C using a wavelength of 0.98 Å at 100 K. Diffraction data sets were indexed, integrated and scaled using XDS and XSCALE (*Kabsch, 2010*). The structure was automatically solved via the Auto-Rickshaw webserver (*Panjikar et al., 2009*), manually adjusted in Coot (*Emsley and Cowtan, 2004*), and further refined with phenix.refine (*Afonine et al., 2012*). The Ramachandran statistics for the

refined model is 98.76% (favored), 1.24% (allowed) and 0 (outliers). Detailed data collection and structure refinement statistics are in *Table 3*.

## Expression and purification of the PCDH15-LHFPL5 complex

HEK293 tsa201 cells at a density of $2.0 \times 10^6$ $ml^{-1}$ were co-infected with either $PCDH15_{4EC}$ and LHFPL5, or $PCDH15_{1EC}$ and LHFPL5 BacMam viruses (*Goehring et al., 2014*) at a MOI of 1:1. Cultures were supplemented with 10 mM sodium butyrate 12 hr post-infection and transferred to 30°C. Cells were harvested 60 hr post-infection and lysed in buffer containing 100 mM Tris pH 8.0, 150 mM NaCl, 1% (w/v) digitonin and protease inhibitors for 2 hr at 4°C. The solubilized material was incubated with Strep-Tactin resin, washed with buffer A containing 20 mM Tris pH 8.0, 150 mM NaCl, 0.07% (w/v) digitonin and eluted with buffer A plus 5 mM desthiobiotin to remove free His-tagged $PCDH15_{4EC}$ or $PCDH15_{1EC}$. The elution was then incubated with TALON resin, washed with buffer A plus 10 mM imidazole to remove free strep-tagged LHFPL5 and further eluted with buffer A plus 200 mM imidazole. After the two-step affinity purification, the $PCDH15_{4EC}$-LHFPL5 or $PCDH15_{1EC}$-LHFPL5 complex was treated with thrombin 1:200 (w/w) to remove strep-CFP fused to LHFPL5 and mVenus-8His fused to PCDH15, and further purified by size-exclusion chromatography (Superose 6 Increase 10/300 GL column) in buffer A. Peak fractions were collected and concentrated for cryo-EM analysis. In the case of $PCDH15_{4EC}$-LHFPL5, 1 mM $CaCl_2$ was included in all buffers.

## AUC analysis

Sedimentation velocity (SV) AUC experiments were carried out in an Optima XL-I analytical ultracentrifuge (Beckman Coulter, Indianapolis, IN) using standard methods (*Zhao et al., 2013*; *Schuck, 2017*). Briefly, samples were loaded into AUC cell assemblies with Epon centerpieces with 12 mm or 3 mm optical path length at a volume to generate 12 mm solution columns. To achieve chemical and thermal equilibrium, the An-50 TI rotor with loaded samples was rested for 2 hr at 20°C in the rotor chamber. After acceleration to 50,000 rpm, continuous absorbance data acquisition at 280 nm was started. SEDFIT (sedfitsedphat.nibib.nih.gov) was used to calculate the diffusion-deconvoluted sedimentation coefficient distribution c(s) (*Schuck, 2000*), which were then loaded into GUSSI (*Brautigam, 2015*) for plotting and integration from 1 to 5.5 S to determine the signal-weighted average sedimentation coefficient $s_w$ at each concentration. The resulting $s_w$ isotherm was analyzed in SEDPHAT (sedfitsedphat.nibib.nih.gov) with a homo-dimerization model allowing for the onset of hydrodynamic nonideality. Monomer and dimer sedimentation coefficients and the equilibrium dissociation constant $K_D$ were refined by non-linear regression, with the non-ideality coefficient fixed at 0.009 mL/mg. To determine the dimer dissociation rate constant $k_{off}$, SV data from samples at 2.3, 7.0 and 11.8 µM were globally analyzed in SEDPHAT by direct boundary modeling (*Schuck, 2000*). $k_{off}$ along with $K_D$ was refined, while the sedimentation coefficients were fixed at the best-fit values from the isotherm analysis. Confidence limits were determined using the error projection method and F-statistics (*Johnson, 1992*) as implemented in SEDPHAT.

## PCDH15-LHFPL5 complex cryo-EM data collection

A 2.5 µL aliquot of the $PCDH15_{4EC}$-LHFPL5 (1.5 mg/ml) or $PCDH15_{1EC}$-LHFPL5 (10 mg/ml) complex was applied to a glow-discharged Quantifoil holey carbon grid (1.2/1.3 µm size/home space, 300 mesh), blotted using a Vitrobox Mark III with blotting time of 4 s, blotting force of 1, 100% humidity and plunge-frozen into a liquid ethane-propane mixture (0.4:0.6, ethane: propane) cooled by liquid nitrogen.

Images of the $PCDH15_{4EC}$-LHFPL5 complex were taken with an FEI Titan Krios electron microscope operating at 300 keV with a nominal magnification of 81 k, and were recorded by a Gatan K2 Summit direct electron detector in super-resolution counting mode with a binned pixel size of 1.7 Å. Each movie stack was dose-fractionated to 50 frames with a total exposure time of 10 s at a defocus range of 0.3–1.3 µm, resulting in a total dose of 27 e/Å2. Contrast was improved by using a Volta phase plate (*Danev and Baumeister, 2016*). During data collection the spot on the phase plate was advanced every 60 min, after which the phase shift reached approximately 120°.

Images of the $PCDH15_{1EC}$-LHFPL5 complex were taken with an FEI Titan Krios electron microscope operating at 300 keV with a nominal magnification of 105 k and were recorded by a Gatan K2 Summit direct electron detector in super-resolution counting mode with a binned pixel size of 1.04

Å. Each movie stack was dose-fractionated to 50 frames with a total exposure time of 10 s at a defocus range of 0.7–2.2 μm, resulting in a total dose of 74 e/Å (*Kazmierczak et al., 2007*).

## PCDH15$_{1EC}$-LHFPL5 complex cryo-EM data processing

Super-resolution movie stacks were 2 × 2 down sampled and motion-corrected using Unblur (*Grant and Grigorieff, 2015*). The contrast transfer function (CTF) parameters were determined by Gctf (*Zhang, 2016*). Images (4,527) with appropriate ice thickness and particle distribution were selected after manual inspection. A total of 972,563 particles were auto-picked with Gautomatch (http://www.mrc-lmb.cam.ac.uk/kzhang/Gautomatch/Gautomatch_v0.56/). Two rounds of 2D classification were carried out to clean up the data set, yielding 736,152 particles. An ab-initio 3D reconstruction was generated using cryoSPARC (*Punjani et al., 2017*), low-pass filtered to 60 Å and used as the reference model for 3D auto-refine with global research and C2 symmetry in RELION (*Scheres, 2012*). C2 symmetry was used in the following processes. The refined particles were subjected to 3D local classification into four classes with a soft mask. The CTF values of individual particles from the best class (244K particles) were estimated using Gctf, followed by 3D auto-refine with local search. The refined particles were further classified into four classes with no alignment in RELION. One class showed obvious better features in the β-sheet area of LHFPL5 and transmembrane helices and was selected for final refinement. The final refinement was initially carried out in RELION with a soft mask around the complex and micelle, resulting in a 4.5 Å reconstitution postprocessing. After masking out the micelle, particles were further refined using manual refinement in cisTEM (*Grant et al., 2018*). The final resolution was based on the gold standard FSC 0.143 criteria and the local resolution was estimated with blocalres program in the bsoft package (*Heymann and Belnap, 2007*). C1 symmetry was tested in processes of 3D classification and refinement, resulting in similar features but less well defined maps compared to those with C2 symmetry, thus helping to justify the application of C2 symmetry.

## PCDH15$_{4EC}$-LHFPL5 complex electron microscopy data processing

Super-resolution movie stacks were 2 × 2 down sampled and motion-corrected using motioncor2 (*Zheng et al., 2017*). The contrast transfer function (CTF) parameters and phase shift were determined by Gctf (*Zhang, 2016*). A total of 2348 images with appropriate ice thickness and particle distribution were selected after manual inspection. Particles were initially picked using DoG-picker (*Voss et al., 2009*). After 2D classification, representative class averages were used as templates for particle picking using Gautomatch. Particles were subjected to two rounds of 2D-classification, after which only particles belonging to classes with clear features were used for subsequent steps. An initial model was calculated using cryoSPARC (*Punjani et al., 2017*) with particles belonging to 2D-classes that clearly showed a 'straight' conformation. This model was used for 3D-classification using RELION. A soft mask and no symmetry was used during classification. The class showing the clearest features and an apparent C2 symmetry was used for auto-refinement in RELION, which was performed using either C2 or C1 symmetry. Resolution of the map was estimated by gold-standard FSC using the RELION postprocessing program and local resolution was estimated using the blocalres program in the bsoft package (*Heymann and Belnap, 2007*). Because the reconstruction using C2 showed similar features to the C1 reconstruction and had a slightly higher resolution it was used for further analysis.

## PCDH15$_{4EC}$-LHFPL5 complex model building

To construct a model of the extracellular domains of PCDH15$_{4EC}$ we used the crystal structure of EC8-EC10 (PDB code: 4XHZ) and our crystal structure of EC11-EL as templates. EC8-9, EC10, and EC11-EL were fit as rigid bodies into the density using Chimera (*Pettersen et al., 2004*). Rosetta-CM (*Song et al., 2013*) then was used to refine the fit and model the linker between the domains. The GlyProt server (*Bohne-Lang and von der Lieth, 2005*) was used to model the common core pentasaccharide Man$_3$GlcNac$_2$ onto asparagine residues that are predicted to be modified. The glycosylations at EC10 and EC11 had clearly corresponding density features, while glycosylations at EC8 did not have corresponding density features and were therefore not included in the model. The CHARMM-Gui server (*Jo et al., 2008*) was used to create a topology file for NAMD (*Phillips et al., 2005*), followed by MDFF refinement of the full-length glycosylated structure into the density

(*Trabuco et al., 2008*). After this fitting pentasaccharide structures were replaced by the lowest energy conformation as suggested by the GlyProt server. For final optimization of geometry, phenix. real_space_refine (*Headd et al., 2012*) was used for global minimization and B-factor refinement using a weighting factor of 0.0001 for electron density to idealize geometry.

## PCDH15$_{1EC}$-LHFPL5 complex model building

The crystal structure of PCDH15 EC11-EL was rigid-body fit into the cryo-EM map using Chimera (*Pettersen et al., 2004*). The orientation of the C-terminal extended helix of the EL domain was adjusted by rigid-body fitting into the density. Density belonging to LHFPL5 was identified by rigid-body fitting the crystal structure of claudin-15 (PDB code: 4P79). The PCDH15$_{1EC}$ helix was constructed as an ideal helix and rigid-body fitted into the remaining density. An initial PCDH15$_{1EC}$-LHFPL5 model was constructed using Rosetta-CM (*Song et al., 2013*) using the initial model of PCDH15$_{1EC}$ and the crystal structures of claudin-4 (PDB code: 5B2G), claudin-15 (PDB code: 4P79), and calcium voltage-gated channel auxiliary subunit gamma 1 (PDB code: 5GJV) as templates. The best scoring models were manually inspected and the most reasonable model with a cross correlation coefficient of 0.815 (0.848 for backbone atoms) was used as a starting model. Tyr1377 in PCDH15$_{1EC}$ showing clear side chain density was used as a marker of the PCDH15$_{1EC}$ transmembrane helix register, together with Pro1395, which is located at the slight bend of the TM helix. Trp29, Trp132, and Trp177, each with substantial side chain densities, were used to inform the register of LHFPL5 transmembrane helices 1, 3, and 4, respectively. A disulfide bond composed of Cys114 on TM2 and Cys130 on TM3 was used to inform the register of LHFPL5 TM2. The densities corresponding to the extracellular region of LHFPL5 were not clear enough to show individual β-strands. The extracellular region of the crystal structures of claudin-15 and claudin-19 were then used to help guide the β-strand orientation and location. Individual β-strands of claudin-15 were rigid-body fitted into the density, mutated to LHFPL5 sequences based on multiple sequence alignments and manually refined in Coot (*Emsley and Cowtan, 2004*). A highly conserved disulfide bond, which also exists in all templates, was built between Cys68 on β3 and Cys79 on β4. The linkers between each β-strand were also manually built and refined in Coot. The loop between TM3 and the fifth β-strand of LHFPL5 (Asp153-Leu170) as well as the loop between the first and second β-strand (Ser55-Pro57) were not built due to lack of side chain densities in the region. Due to weak density, the terminal residues Val2-Tyr14 and Gly200-Val219 of LHFPL5 and residues Gln1411-His1462 of PCDH15$_{1EC}$ were not included in the model. Final refinement was performed using phenix.real_space_refine (*Headd et al., 2012*) with secondary structure restraints, resulting in a final cross correlation of 0.837 (0.875 for backbone atoms).

## Cysteine crosslinking

Residues from Glu1373 to Ala1386, with the exception of Leu1375, were individually mutated to cysteine in the background of the PCDH15$_{4EC}$ Δ1413 construct using the Quikchange mutagenesis kit. Adherent HEK293 tsa201 cells were co-transfected with the indicated cysteine mutant and LHFPL5. After 8 hr the media was exchanged to media containing 10 mM sodium butyrate and temperature shifted to 30°C. After 60 h cells were harvested and lysed with lysis buffer (1% digitonin, 150 mM NaCl, 50 mM Tris pH 8.0, 1 mM CaCl$_2$). Following 2 hr agitation at 4°C the lysates were clarified by ultracentrifugation and then mixed with 4x SDS loading buffer containing either 40 mM *N*-ethylmaleimide or 500 mM β-mercaptoethanol for oxidizing and reducing conditions, respectively. The samples were separated by SDS-PAGE and imaged using a fluorescence imager.

## Acknowledgements

The authors would like to thank Teresa Nicolson and Peter Barr-Gillespie for comments on the experiments associated with this manuscript and on the manuscript itself. Cryo-EM data has been collected at the OHSU Multiscale Microscopy Core and at the HHMI cryo-EM facility at Janelia campus. X-ray diffraction data was collected at ALS beamline 8.2.1 and APS beamline 24-ID-C. This work was supported by the intramural research program of NIBIB, National Institutes of Health, Bethesda, U.S.A. E.G. is an investigator with the Howard Hughes Medical Institute. The NE-CAT beamlines are funded by the NIH (NIGMS grant P41 GM103403 and NIH-ORIP HEI grant S10 RR029205), and uses

resources of the Advanced Photon Source at Argonne National Laboratory under DOE Contract No. DE-AC02-06CH11357.

## Additional information

### Funding

| Funder | Grant reference number | Author |
|---|---|---|
| National Institutes of Health | NIBIB intramural research program | Peter Schuck Huaying Zhao |
| Howard Hughes Medical Institute | | Eric Gouaux |

The funders had no role in study design, data collection and interpretation, or the decision to submit the work for publication.

### Author contributions

Jingpeng Ge, Johannes Elferich, Conceptualization, Data curation, Formal analysis, Validation, Investigation, Visualization, Methodology, Writing—original draft, Writing—review and editing; April Goehring, Data curation, Formal analysis, Writing—review and editing; Huaying Zhao, Formal analysis, Investigation, Visualization, Methodology, Writing—review and editing; Peter Schuck, Formal analysis, Funding acquisition, Investigation, Visualization, Methodology, Writing—review and editing; Eric Gouaux, Conceptualization, Resources, Formal analysis, Supervision, Funding acquisition, Methodology, Writing—original draft, Project administration, Writing—review and editing

### Author ORCIDs

Jingpeng Ge (ID) http://orcid.org/0000-0001-6164-1221
Johannes Elferich (ID) http://orcid.org/0000-0002-9911-706X
April Goehring (ID) http://orcid.org/0000-0002-2592-6956
Huaying Zhao (ID) http://orcid.org/0000-0002-8827-6639
Peter Schuck (ID) http://orcid.org/0000-0002-8859-6966
Eric Gouaux (ID) http://orcid.org/0000-0002-8549-2360

### Decision letter and Author response

Decision letter https://doi.org/10.7554/eLife.38770.033
Author response https://doi.org/10.7554/eLife.38770.034

## Additional files

### Supplementary files

• Transparent reporting form
DOI: https://doi.org/10.7554/eLife.38770.021

### Data availability

The crystal structure of EC11-EL has been deposited to the Protein Data Bank under accession code 6C10. The three-dimensional cryo-EM density maps of the PCDH15$_{4EC}$-LHFPL5 complex and the PCDH15$_{1EC}$-LHFPL5 complex have been deposited to the EM database under the accession codes EMD-7327 and EMD-7328, respectively, and the coordinates for the structures have been deposited to the Protein Data Bank under the accession codes 6C13 and 6C14, respectively.

The following datasets were generated:

| Author(s) | Year | Dataset title | Dataset URL | Database, license, and accessibility information |
|---|---|---|---|---|
| Eric Gouaux, Johannes Elferich, | 2018 | Crystal structure of mouse PCDH15 EC11-EL | http://www.rcsb.org/structure/6C10 | Publicly available at the RCSB Protein |

| Jingpeng Ge | | | | Data Bank (accession no: PDB 6C10). |
|---|---|---|---|---|
| Eric Gouaux, Johannes Elferich, Jingpeng Ge | 2018 | CryoEM structure of mouse PCDH15-4EC-LHFPL5 complex | http://www.rcsb.org/structure/6C13 | Publicly available at the RCSB Protein Data Bank (accession no: PDB 6C13). |
| Eric Gouaux, Johannes Elferich, Jingpeng Ge | 2018 | CryoEM structure of mouse PCDH15-4EC-LHFPL5 complex | http://www.ebi.ac.uk/pdbe/entry/emdb/EMD-7327 | Publicly available at the Electron Microscopy Data Bank (accession no: EMD-7327). |
| Eric Gouaux, Johannes Elferich, Jingpeng Ge | 2018 | CryoEM structure of mouse PCDH15-1EC-LHFPL5 complex | http://www.rcsb.org/structure/6C14 | Publicly available at the RCSB Protein Data Bank (accession no: PDB 6C14). |
| Eric Gouaux, Johannes Elferich, Jingpeng Ge | 2018 | CryoEM structure of mouse PCDH15-1EC-LHFPL5 complex | http://www.ebi.ac.uk/pdbe/entry/emdb/EMD-7328 | Publicly available at the Electron Microscopy Data Bank (accession no: EMD-7328). |

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
