## [Decision Letter]

Thank you for submitting your article "Structure of protocadherin 15 of the stereocilia 14 tip link in complex with LHFPL5" for consideration by *eLife*. Your article has been reviewed by three peer reviewers, and the evaluation has been overseen by a Reviewing Editor and Richard Aldrich as the Senior Editor. The following individuals involved in review of your submission have agreed to reveal their identity: Lawrence Shapiro (Reviewer #1); Anthony J Ricci (Reviewer #2); Gregory M Alushin (Reviewer #3).

The reviewers have discussed the reviews with one another and the Reviewing Editor has drafted this decision to help you prepare a revised submission.

Summary:

This is an interesting and important piece of work that provides structural evidence for how two components of the tip link macromolecular structure interact and how these interactions might play into force transfer due to hair bundle motion. Using heterologous co-expression in mammalian cells and FSEC analysis, the authors identify a stable subcomplex of PCDH15 and LHFPL5, a putative component of the MET channel, with no direct interaction detected between PCDH15 and two other putative MET components (TMC1 and TMIE). They then perform an extensive structural analysis of this complex using a combined approach of cryo-EM of stable regions and X-ray crystallography of key domains, as well as supporting AUC and cysteine cross-linking experiments. In sum, their experiments identify a long, flexible PCDH shaft linked to a stable EC11-EL collar, which could transduce mechanical forces applied to the shaft into separation of transmembrane helices, potentially facilitating gating of the MET channel. This is a great start for a unique and critical channel that has resisted prior attempts at molecular characterization.

Essential revisions:

- In Figure 2—figure supplement 2 and the Results section, the authors claim that separation of β-strands and a well-defined α-helix in the density map validates their docking. These features should not be visible in an 11 Å resolution map: the fit should rather be guided by shape complimentarity of rigid-body docked domains (which is what the authors actually do). Furthermore, there is no visible α-helical density (or an α-helix in the model) in Figure 2—figure supplement 2G. It is recommended to remove this from the text.

- For all their model-map comparisons, the authors employ a FSC 0.25 criterion. There is no statistical basis for choosing this criterion: an FSC 0.5 is appropriate for model-map comparisons. While this criterion was introduced by Sjors Scheres and colleagues in 2015, Scheres recently recanted on the 3DEM message board. Please don't perpetuate this, unless the authors have a specific justification.

- The Discussion and Figure 6 would benefit from a discussion of the implications of the tilted conformations of the PCDH15 rod relative to the transmembrane/collar regions, a major observation which goes unmentioned in the model. The interpretation of the data should be left to the authors, but a few possibilities can be envisioned. One would be that tilting of the rod could also be an activation mechanism. Another would be that tilting encodes a directional preference to activation, as has been observed by Hudspeth and colleagues (e.g. the rod tilts if pulled in the wrong direction, preventing inappropriate channel opening). Obviously, the authors' data don't strongly support any model over another at this stage, but it could stimulate future work to acknowledge more possibilities than the straight conformation being the relevant one for activation.

- Stated in the Abstract as well as the Discussion is that new light is shed upon how force from PCDH15 is translated to the MET channel. The data provided do not show this specifically but lead to a proposed model. Some re-wording would be useful here.

- The text in the second paragraph of the Introduction is a bit overstated and not particularly complete regarding the current state of knowledge. There is considerable debate over Myosin 7a or Myosin Ic. There is no clear data about how the MET channel is coupled to PCDH15. If the argument is that LHFPL5 is part of the MET channel that should be spelled out a bit more clearly.

- Are there concerns that interactions identified from the heterologous expression system truly mimic what happens in vivo? For example, are there differences in post-translational modifications that could alter interactions?

- Subsection “Cryo-EM structure of PCDH15_1EC_-LHFPL5”, third paragraph: Please comment on whether the extensive interactions within the TM region could be affected by the lipid environment or altered by the additional TM proteins found in this region.

*Reviewer #1:*

This manuscript, by Gouaux and colleagues, addresses a complex question of enormous importance – the architecture of the mechano-electrical transduction (MET) ion channel attached to the PCDH15 component of the tip link. As the authors describe, while the overall architecture of the system for hearing perception is well understood, including the existence of a PCDH15-attached MET channel whose open probability is modulated by tension of the tip link, which in turn is modulated by auditory deflection of hair cell stereocilia, molecular knowledge of the system is sparse. While numerous putative components of the hair cell MET channel have been identified, these components have not been successfully reconstituted and the architecture of the MET complex remains unknown.

Here, Gouaux and colleagues begin to reconstitute the system through which auditory perception is mediated. Beginning with an extracellularly truncated PCDH15, the authors screen putative channel components, identifying a stable PCDH15/LHFPL5 complex, which they characterize and reconstruct by cryo-EM. This PCDH15 construct contains a non-cadherin "EL" domain downstream from EC11. They produce a high-resolution structure of this fragment, revealing the EL domain to form a collar around the dimeric PCDH15 ectodomain near the membrane insertion point. This high-resolution structure is used in interpretation of the cryo-EM reconstruction of the PCDH15 complex with LHFPL5, revealing two molecules of the claudin-like LHFPL5 to act as a clamp on the dimeric PCDH15 within the membrane.

Overall this is an excellent paper. While the results do not reveal the architecture of the MET channel, they do reveal the first "layer" of association, providing a jumping off point to assemble the rest of the channel. This is a great start for a unique and critical channel that has resisted prior attempts at molecular characterization. Overall, this is an excellent contribution, highly appropriate for publication in *eLife*.

*Reviewer #2:*

This is an interesting and important piece of works that provides structural evidence for how two components of the tip link macromolecular structure interact and how these interactions might play into force transfer due to hair bundle motion. My comments are relatively minor and go largely at physiological relevance. These include:

1) Stated in the Abstract as well as the Discussion is that new light is shed upon how force from PCDH15 is translated to the MET channel. I don't see how data provided does this specifically. At best the work has implications for how force might be translated across the membrane but unless we know how the channel is integrated into the complex we cannot say how the force is presented to the channel can we?

2) The text in the second paragraph of the Introduction is a bit overstated about what we know and not particularly complete. There is considerable debate over Myosin 7a or Myosin Ic. There is no clear data about how the MET channel is coupled to PCDH15. If the argument is that LHFPL5 is part of the MET channel that should be spelled out a bit more clearly.

3) How do you know the interactions identified using in vitro pull down experiments are the same as what happens in vivo? Does the expression system matter? Are there differences post-translationally that could alter interactions?

4) Although TMC1 was include as part of the pulldowns, the data was not discussed? How are these data interpreted? Is the interaction between PCDH15 and LHFPL5 altered by addition of other known proteins? Is it altered by the specific lipid environment that it was prepared in?

5) Subsection “Cryo-EM structure of PCDH15_4EC_-LHFPL5”, end of last paragraph – probably best to leave speculation in the Discussion, as you don't know the tension required to move between conformations or that the difference between states is tension dependent (although it might be) better to describe the idea along with the model diagrams at the end.

6) Subsection “Crystal structure of PCDH15 EC11-EL”, third paragraph – is there a Figure 5—figure supplement 1F?

7) Subsection “Cryo-EM structure of PCDH15_1EC_-LHFPL5”, third paragraph – how are the extensive interactions within the TM region likely affected by the lipid environment? Do you expect these interactions to be altered by the additional TM proteins found in this region?

*Reviewer #3:*

In their manuscript "Structure of protocadherin 15 of the stereocilia tip link in complex with LHFPL5", Ge, Elferich, and colleagues make an important step forward in defining the biophysical mechanism of mechanotransduction in hair cells, one of the abiding mysteries in this field. Using heterologous co-expression in mammalian cells and FSEC analysis, the authors identify a stable subcomplex of PCDH15 and LHFPL5, a putative component of the MET channel, with no direct interaction detected between PCDH15 and two other putative MET components (TMC1 and TMIE). They then perform an extensive structural analysis of this complex using a combined approach of cryo-EM of stable regions and X-ray crystallography of key domains, as well as supporting AUC and cysteine cross-linking experiments. In sum, their experiments identify a long, flexible PCDH shaft linked to a stable EC11-EL collar, which could transduce mechanical forces applied to the shaft into separation of transmembrane helices, potentially facilitating gating of the MET channel.

While substantial future efforts will be required to define the full MET complex and its gating mechanism, this study makes an important contribution in defining a bona fide subcomplex and describing its detailed structure, which will guide future studies both in vivo and in vitro. Overall, the experimental work is extremely thorough and, to this reviewer, convincing. I believe this story adds up to a satisfying advance, and I think it is suitable for publication in *eLife* pending the address of minor concerns outlined below, without further experimental work.

Minor concerns:

- In Figure 2—figure supplement 2 and the Results section, the authors claim that separation of β-strands and a well-defined α-helix in the density map validates their docking. These features should not be visible in an 11 Å resolution map: the fit should rather be guided by shape complimentarity of rigid-body docked domains (which is what the authors actually do). Furthermore, I don't see any α-helical density (or an α-helix in the model) in Figure 2—figure supplement 2G. I recommend simply removing this from the text.

- For all their model-map comparisons, the authors employ a FSC 0.25 criterion. There is no statistical basis for choosing this criterion: an FSC 0.5 is appropriate for model-map comparisons. While this criterion was introduced by Sjors Scheres and colleagues in 2015, Scheres recently recanted on the 3DEM message board. Please don't perpetuate this, unless the authors have a specific justification I am unaware of.

- I believe the Discussion and Figure 6 would benefit from a discussion of the implications of the tilted conformations of the PCDH15 rod relative to the transmembrane/collar regions, a major observation which goes unmentioned in the model. I leave interpretation of their data to the authors, but I can envision a few possibilities. One would be that tilting of the rod could also be an activation mechanism. Another would be that tilting encodes a directional preference to activation, as has been observed by Hudspeth and colleagues (e.g. the rod tilts if pulled in the wrong direction, preventing inappropriate channel opening). Obviously, the authors' data don't strongly support any model over another at this stage, but I do think it could stimulate future work to acknowledge more possibilities than the straight conformation being the relevant one for activation.

---

## [Author Response]

Essential revisions:- In Figure 2—figure supplement 2 and the Results section, the authors claim that separation of β-strands and a well-defined α-helix in the density map validates their docking. These features should not be visible in an 11 Å resolution map: the fit should rather be guided by shape complimentarity of rigid-body docked domains (which is what the authors actually do). Furthermore, there is no visible α-helical density (or an α-helix in the model) in Figure 2—figure supplement 2G. It is recommended to remove this from the text.

We agree and have removed the statement and have revised the text. The revised text is “We were unable to fit this conformation of EC9-EC10 into our density map and therefore fitted EC8, EC9, and EC10 as independent rigid bodies, guided by the shape complementarity of the rigid-body docked domains.”

- For all their model-map comparisons, the authors employ a FSC 0.25 criterion. There is no statistical basis for choosing this criterion: an FSC 0.5 is appropriate for model-map comparisons. While this criterion was introduced by Sjors Scheres and colleagues in 2015, Scheres recently recanted on the 3DEM message board. Please don't perpetuate this, unless the authors have a specific justification.

We thank the reviewer for pointing it out. We revised all the model-map comparisons using a FSC 0.5 criterion.

- The Discussion and Figure 6 would benefit from a discussion of the implications of the tilted conformations of the PCDH15 rod relative to the transmembrane/collar regions, a major observation which goes unmentioned in the model. The interpretation of the data should be left to the authors, but a few possibilities can be envisioned. One would be that tilting of the rod could also be an activation mechanism. Another would be that tilting encodes a directional preference to activation, as has been observed by Hudspeth and colleagues (e.g. the rod tilts if pulled in the wrong direction, preventing inappropriate channel opening). Obviously, the authors' data don't strongly support any model over another at this stage, but it could stimulate future work to acknowledge more possibilities than the straight conformation being the relevant one for activation.

We agree that the role of the tilted conformation should be discussed in the manuscript, even though our data cannot address the question of physiological significance. We have updated Figure 6 and have added the following paragraph to the Discussion:

“The two linker regions connecting the PCDH15 collar with the TM domains might act as “ligaments” of a hinge joint, allowing the tilting of the PCDH15 extracellular domains. […] These conformational changes could ultimately lead to the gating of MET ion channel.”

- Stated in the Abstract as well as the Discussion is that new light is shed upon how force from PCDH15 is translated to the MET channel. The data provided do not show this specifically but lead to a proposed model. Some re-wording would be useful here.

Text was revised to “and shed new light on how forces in the PCDH15 tether may be transduced into the stereocilia membrane.”

- The text in the second paragraph of the Introduction is a bit overstated and not particularly complete regarding the current state of knowledge. There is considerable debate over Myosin 7a or Myosin Ic. There is no clear data about how the MET channel is coupled to PCDH15. If the argument is that LHFPL5 is part of the MET channel that should be spelled out a bit more clearly.

We agree with the reviewer and have removed the relevant sentence.

- Are there concerns that interactions identified from the heterologous expression system truly mimic what happens in vivo? For example, are there differences in posttranslational modifications that could alter interactions?

Please see reply to comment below.

- Subsection “Cryo-EM structure of PCDH15_1EC_-LHFPL5”, third paragraph: Please comment on whether the extensive interactions within the TM region could be affected by the lipid environment or altered by the additional TM proteins found in this region.

We agree with the reviewers that the unique environment of hair cells might affect this complex and our structure therefore might not represent the lower tip link in vivo. However, the low abundance of hair cells makes it difficult to work with native material and therefore a mammalian cell culture system, like that used here, is in our opinion a reasonable compromise. We have added the following sentence to the Discussion to make these concerns clear:

“Because the material obtained for structure determination is not purified from hair cells we cannot exclude the possibility that the unique cellular architecture, proteome (Krey et al., 2017) or lipid composition (Zhao et al., 2012) of hair cells could influence the conformation of the complex and the observed interactions between the protein components.”

Reviewer #1:

[…] Overall this is an excellent paper. While the results do not reveal the architecture of the MET channel, they do reveal the first "layer" of association, providing a jumping off point to assemble the rest of the channel. This is a great start for a unique and critical channel that has resisted prior attempts at molecular characterization. Overall, this is an excellent contribution, highly appropriate for publication in eLife.

Reviewer #2:This is an interesting and important piece of works that provides structural evidence for how two components of the tip link macromolecular structure interact and how these interactions might play into force transfer due to hair bundle motion. My comments are relatively minor and go largely at physiological relevance. These include:1) Stated in the Abstract as well as the Discussion is that new light is shed upon how force from PCDH15 is translated to the MET channel. I don't see how data provided does this specifically. At best the work has implications for how force might be translated across the membrane but unless we know how the channel is integrated into the complex we cannot say how the force is presented to the channel can we?

We have revised the text as described previously.

2) The text in the second paragraph of the Introduction is a bit overstated about what we know and not particularly complete. There is considerable debate over Myosin 7a or Myosin Ic. There is no clear data about how the MET channel is coupled to PCDH15. If the argument is that LHFPL5 is part of the MET channel that should be spelled out a bit more clearly.

Revised as suggested.

3) How do you know the interactions identified using in vitro pull down experiments are the same as what happens in vivo? Does the expression system matter? Are there differences post translationally that could alter interactions?

We have revised the text as previously described.

4) Although TMC1 was include as part of the pulldowns, the data was not discussed? How are these data interpreted? Is the interaction between PCDH15 and LHFPL5 altered by addition of other known proteins? Is it altered by the specific lipid environment that it was prepared in?

The pulldowns with TMC1 resulted in heterogeneous samples that were not further characterized. The interaction that we observed between PCDH15 and LHFPL5 was not perceptibly altered by the expression of other known proteins. We experimented with many different lipids and detergents for solubilization and did not find any mixtures that discernably altered the nature of the pulldowns and complexes. Full presentation of these experiments and their interpretation is beyond the scope of the present manuscript.

5) Subsection “Cryo-EM structure of PCDH15_4EC_-LHFPL5”, end of last paragraph – probably best to leave speculation in the Discussion, as you don't know the tension required to move between conformations or that the difference between states is tension dependent (although it might be) better to describe the idea along with the model diagrams at the end.

Revised. The sentence was removed from the Results section.

6) Subsection “Crystal structure of PCDH15 EC11-EL”, third paragraph – is there a Figure 5—figure supplement 1F?

Revised. The third paragraph of the subsection “Crystal structure of PCDH15 EC11-EL” refers to Figure 5—figure supplement 1A.

7) Subsection “Cryo-EM structure of PCDH15_1EC_-LHFPL5”, third paragraph – how are the extensive interactions within the TM region likely affected by the lipid environment? Do you expect these interactions to be altered by the additional TM proteins found in this region?

These studies are beyond the scope of the present manuscript.

Reviewer #3:[…] Minor concerns:- In Figure 2—figure supplement 2 and the Results section, the authors claim that separation of β-strands and a well-defined α-helix in the density map validates their docking. These features should not be visible in an 11 Å resolution map: the fit should rather be guided by shape complimentarity of rigid-body docked domains (which is what the authors actually do). Furthermore, I don't see any α-helical density (or an α-helix in the model) in Figure 2—figure supplement 2G. I recommend simply removing this from the text.

Revised.

- For all their model-map comparisons, the authors employ a FSC 0.25 criterion. There is no statistical basis for choosing this criterion: an FSC 0.5 is appropriate for model-map comparisons. While this criterion was introduced by Sjors Scheres and colleagues in 2015, Scheres recently recanted on the 3DEM message board. Please don't perpetuate this, unless the authors have a specific justification I am unaware of.

Comparison revised as suggested.

- I believe the Discussion and Figure 6 would benefit from a discussion of the implications of the tilted conformations of the PCDH15 rod relative to the transmembrane/collar regions, a major observation which goes unmentioned in the model. I leave interpretation of their data to the authors, but I can envision a few possibilities. One would be that tilting of the rod could also be an activation mechanism. Another would be that tilting encodes a directional preference to activation, as has been observed by Hudspeth and colleagues (e.g. the rod tilts if pulled in the wrong direction, preventing inappropriate channel opening). Obviously, the authors' data don't strongly support any model over another at this stage, but I do think it could stimulate future work to acknowledge more possibilities than the straight conformation being the relevant one for activation.

Revised as described above.